# METACLUSTER: ENABLING DEEP COMPRESSION OF KOLMOGOROV-ARNOLD NETWORK

## ABSTRACT

Kolmogorov-Arnold Networks (KANs) replace scalar weights with per-edge vectors of basis coefficients, thereby boosting expressivity and accuracy but at the same time resulting in a multiplicative increase in parameters and memory. We propose MetaCluster, a framework that makes KANs highly compressible without sacrificing accuracy. Specifically, a lightweight meta-learner, trained jointly with the KAN, is used to map low-dimensional embedding to coefficient vectors, shaping them to lie on a low-dimensional manifold that is amenable to clustering. We then run K-means in coefficient space and replace per-edge vectors with shared centroids. Afterwards, the meta-learner can be discarded, and a brief fine-tuning of the centroid codebook recovers any residual accuracy loss. The resulting model stores only a small codebook and per-edge indices, exploiting the vector nature of KAN parameters to amortize storage across multiple coefficients. On MNIST, CIFAR-10, and CIFAR-100, across standard KANs and ConvKANs using multiple basis functions, MetaCluster achieves a reduction of up to $80\times$ in parameter storage, with no loss in accuracy. Similarly, on high-dimensional equation modeling tasks MetaCluster achieves a parameter reduction of $124.1\times$, without impacting performance. Code will be released upon publication.

## 1 INTRODUCTION

Kolmogorov–Arnold Networks (KANs) have recently emerged as a compelling alternative to multi-layer perceptrons (MLPs), delivering strong results in equation modeling and scientific machine learning, often with improved task performance at comparable or lower parameter counts in those settings (Liu et al., 2024b; Li et al., 2025; Coffman & Chen, 2025; Koenig et al., 2024). Recently, KANs have also begun to show promise in computer vision (Yang & Wang, 2024; Raffel & Chen, 2025). However, unlike equation modeling, at larger scales, KANs frequently incur a substantial parameter overhead relative to MLPs (Yu et al., 2024). This overhead stems from KANs per-edge degrees of freedom: each connection carries a vector of basis coefficients (e.g., B-spline weights) rather than a single scalar weight, resulting in a multiplicative increase in parameters.

One approach that targets this multiplicative increase in parameters is weight sharing, which clusters parameters into a codebook that stores compact indices. Unfortunately, naively applying weight sharing to KANs is ineffective. Instead of clustering scalars (as in MLPs), we must cluster high-dimensional coefficient vectors in KANs. In such high-dimensional spaces, absolute distances grow but also concentrate (nearest and farthest become similar). With such an effect brought by the curse of dimensionality, typical clustering methods struggle to form tight clusters (Beyer et al., 1999; Donoho et al., 2000).

We address this challenge with MetaCluster, a three-stage compression framework that merges meta-learning with weight sharing. First, a small meta-learner maps low-dimensional embeddings into per-edge coefficient vectors, constraining KAN activations to a low-dimensional manifold while training on the task loss. This manifold shaping makes the coefficient vectors highly clusterable. Second, we run K-means on the generated coefficients, replacing per-edge weights with codebook centroids indexed with compact codes. Finally, we discard the meta-learner and embeddings, and lightly fine-tune the centroids to recover any accuracy loss. Since each centroid stores an entire coefficient vector, the codebook amortizes over many scalars, yielding a much higher compression factor for KANs than for MLPs at the same number of clusters.

We validate MetaCluster on two model families consisting of a fully-connected KAN (Liu et al., 2024b) and a convolutional KAN (ConvKAN) (Bodner et al., 2024; Drokin, 2024). For each of these models, we test the efficacy of using B-Splines, radial basis functions (RBFs), and Gram polynomials as the bases (Liu et al., 2024b; Li, 2024; SS et al., 2024). To further verify the robustness of our approach, we validate it across MNIST (Deng, 2012), CIFAR-10, CIFAR-100 (Krizhevsky, 2009), and equation modeling. From our experiments, we find that MetaCluster achieves a reduction of up to $80\times$ and $124.1\times$ in parameter storage on image classification and equation modeling relative to the uncompressed KAN without degrading accuracy, is robust across various architectures and datasets, and ablations confirm that enforcing a low-dimensional manifold is key to high-quality clustering.

The main contributions of this paper are:

1. We identify the potential of weight-sharing for reducing the KANs memory footprint and, to our knowledge, provide the first effective weight-sharing method tailored to KANs.

2. We propose a meta-learning approach that shapes per-edge KAN coefficients to lie on a low-dimensional manifold, enabling effective clustering in high dimensions.

3. We provide extensive image classification and equation modeling experiments demonstrating up to $80\times$ and $124.1\times$ memory reduction, respectively, with no loss in accuracy, along with extensive ablations.

## 2 PRELIMINARIES AND MOTIVATION

### 2.1 KOLMOGOROV-ARNOLD NETWORKS

KANs have gained attention as an alternative to conventional MLPs (Liu et al., 2024b). Their design is motivated by the Kolmogorov–Arnold representation theorem, which guarantees that any continuous multivariate function on a bounded domain, $f : [0,1]^n \rightarrow \mathbb{R}$, can be expressed as a finite sum of compositions of univariate continuous functions, $\phi_{q,p} : [0,1] \rightarrow \mathbb{R}$ and $\phi_q : \mathbb{R} \rightarrow \mathbb{R}$ such that

$$f(\mathbf{x}) = f(x_1, ..., x_n) = \sum_{q=1}^{2n+1} \phi_q(\sum_{p=1}^{n} \phi_{q,p}(x_p)). \tag{1}$$

While Equation 1 captures the classical theoretical form, Liu et al. (2024b) demonstrates a practical generalization that permits arbitrary width and depth tailored to the task. We can express such a formulation with $L$ layers as

$$f(\mathbf{x}) = (\Phi_L \circ \Phi_{L-1} \circ ... \circ \Phi_1)\mathbf{x}, \qquad \Phi_l = \begin{bmatrix} \phi_{l,1,1}(\cdot) & \cdots & \phi_{l,1,n_l}(\cdot) \\ \vdots & \ddots & \vdots \\ \phi_{l,n_{l+1},1}(\cdot) & \cdots & \phi_{l,n_{l+1},n_l}(\cdot) \end{bmatrix} \tag{2}$$

In Equation 2, we let $n_l$ denote the number of inputs to layer $l$, with inputs $x_{l,i}$. Each edge from input $i$ to output $j$ in layer $l$ is equipped with a learnable univariate activation $\phi_{l,j,i}(\cdot)$, for $i \in [1, n_l]$ and $j \in [1, n_{l+1}]$.

The most common choice for implementing $\phi_{l,j,i}(\cdot)$ has been through a weighted summation of basis functions $B_i(\cdot)$ represented as,

$$\phi_{l,j,i}(\cdot) = \sum_{i=1}^{|\mathbf{w}|} w_i B_i(\cdot). \tag{3}$$

In Equation 3, the weighted summation of basis functions is parameterized with learnable coefficients, $\mathbf{w} = [w_1, ..., w_{|\mathbf{w}|}]$.

From its increased representation power compared to the MLP, the KAN has offered impressive results on equation modeling tasks for scientific applications (Li, 2024; Li et al., 2025; Coffman & Chen, 2025; Koenig et al., 2024) and has even started to extend its reach into computer vision (Raffel & Chen, 2025; Yang & Wang, 2024). Despite these gains, widespread adoption in modern

large-scale architectures has been hindered by memory inefficiency (Yu et al., 2024). The root cause is structural: each KAN edge carries a vector of basis coefficients (e.g., B-spline weights), whereas an MLP edge carries a single scalar. Ignoring biases, an arbitrary MLP with $L$ layers will possess $\sum_{l=0}^{L-1}(n_l \times n_{l+1})$ parameters, whereas a KAN will possess $\sum_{l=0}^{L-1}(n_l \times n_{l+1}) \times (|\mathbf{w}|)$ parameters. Thus, for identical topologies, a KAN is approximately $|\mathbf{w}|$ times larger than its MLP counterpart.

## 2.2 MOTIVATIONS AND CHALLENGES OF WEIGHT SHARING FOR KAN

These observations motivate a compression strategy that targets the dimensionality of the vector of basis coefficients. Weight sharing is one method that reduces dimensionality directly by clustering parameters into a small codebook and storing compact indices. In its classical form for MLPs, one applies K-means to the set of scalar weights $\mathbf{W} = w_1, w_2, \ldots, w_n$, obtaining centroids $\mathbf{C} = c_1, c_2, \ldots, c_k$ and assignments that minimize within-cluster sum of squares (Han et al., 2015):

$$\arg\min_{\mathcal{C}} \sum_{i=1}^{k} \sum_{w \in C_i} (w - c_i)^2 \qquad (4)$$

Weight sharing has repeatedly been shown to preserve accuracy while substantially reducing the number of parameters (Han et al., 2015; Cho et al., 2021). In MLPs, however, their effectiveness is constrained by the need to store, for every weight, an index that maps back to a codebook centroid; an overhead that limits the ultimate compression. Under a simple model with $n$ weights, a codebook of $k$ centroids stored at $b$ bits per scalar, the achievable compression factor is

$$r = \frac{nb}{n \log_2(k) + kb} \qquad (5)$$

The dominant term in the denominator, $n \log_2 k$, represents the per-weight index cost required to record the mapping from each original weight to its corresponding centroid. When extending weight sharing to KANs, the index mapping is still required, but its relative cost can be significantly reduced because we cluster $|\mathbf{w}|$-dimensional coefficient vectors per edge. Thus, each centroid stores $|\mathbf{w}|$ scalars, amortizing the codebook over many parameters while the index cost remains $n \log_2 k$. We detail this amortization and its impact on compression in Section 3.4.

While weight sharing is promising to save substantial storage for KANs, directly applying it to KANs for model compression is nontrivial. For instance, since each edge on a KAN is defined by $|\mathbf{w}|$ weights rather than 1 as in the MLP, the dimensionality of the points we must cluster is far greater. In increasing the dimensionality, the vector space becomes sparser and points become more equally spaced apart, making clustering points more difficult (Beyer et al., 1999; Donoho et al., 2000). Therefore, to effectively cluster the KAN activations, we require a fundamentally different method for transforming the high-dimensional vector into a more manageable space.

## 3 METHODS

This section presents MetaCluster, a three-stage framework that makes per-edge KAN activation coefficients amenable to clustering and compression. First, we train a single meta-learner that turns compact embeddings into full activation-coefficient vectors, forcing them onto a task-aligned low-dimensional manifold. Next, we run K-means on those vectors and record, for each edge, which centroid it belongs to in that layer. Finally, we discard the meta-learner and embeddings, replace per-edge weights with their assigned centroids via a lookup table, and briefly fine-tune the network to recover any lost accuracy.

### 3.1 MANIFOLD LEARNING THROUGH A META-LEARNER

Clustering the weights of a KAN is challenging due to the high dimensionality of each weight vector. To address this, we introduce a single meta-learner, $M_\theta$, which maps a lower-dimensional embedding $\mathbf{z}_i \in \mathbb{R}^{d_{emb}}$ to the full KAN weights $\mathbf{w}_i \in \mathbb{R}^{|\mathbf{w}|}$, constraining them to lie on a low-dimensional manifold. Formally, the mapping is defined as:

$$M_\theta(\mathbf{z}_i) = \mathbf{W}_2 \sigma(\mathbf{W}_1 \mathbf{z}_i + b_1) + b_2 = \mathbf{w}_i, \qquad (6)$$

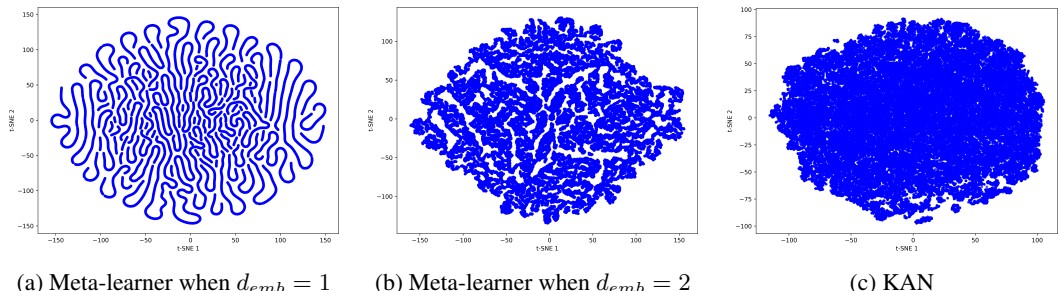

(a) Meta-learner when $d_{emb} = 1$     (b) Meta-learner when $d_{emb} = 2$     (c) KAN

Figure 1: T-SNE visualization of KAN activation weights with and without the meta-learner $M_\theta$. (a) With a 1D embedding, the MetaKAN learns a nearly one-dimensional manifold, (b) with a 2D embedding, it learns a structured two-dimensional manifold, while (c) the baseline KAN without a meta-learner fails to organize the weights into a coherent low-dimensional structure.

where $W_1 \in \mathbb{R}^{d_{\text{hidden}} \times d_{\text{emb}}}$, $W_2 \in \mathbb{R}^{|\mathbf{w}| \times d_{\text{hidden}}}$, and $\sigma(\cdot)$ is a ReLU activation. The meta-learner is trained jointly with the KAN using standard backpropagation for $\alpha$ epochs, so that the generated weights $\mathbf{w_i}$ both lie on a meaningful manifold and optimize the task-specific loss.

To demonstrate the manifold-shaping effect of the meta-learner, we train a two-layer KAN with a B-spline basis on CIFAR-10 Krizhevsky (2009) for one epoch under three settings: (i) a meta-learner with $d_{\text{emb}} = 1$, (ii) a meta-learner with $d_{\text{emb}} = 2$, and (iii) a baseline KAN without a meta-learner. We collect all per-edge activation coefficient vectors $\mathbf{w}_i$ and visualize them with t-SNE. Each point corresponds to $\mathbf{w}_i \in \mathbb{R}^{|\mathbf{w}|}$, where $|\mathbf{w}| = G + k + 1$ with $G = 5$ and $k = 3$, yielding $|\mathbf{w}| = 9$. As shown in Figure 1a, when $d_{\text{emb}} = 1$ the model organizes coefficients along an effectively one-dimensional manifold. Increasing to $d_{\text{emb}} = 2$ produces the well-structured two-dimensional sheet shown in Figure 1b, indicating that the meta-learner concentrates variability onto task-relevant directions. In contrast, the baseline KAN provided in Figure 1c yields a diffuse cloud without coherent low-dimensional structure, which hampers downstream clustering. These visualizations substantiate our premise: shaping coefficient vectors via a low-dimensional embedding makes them markedly more clusterable, a property we later translate into the higher compression at fixed accuracy shown in Section 4.2.

## 3.2 KAN ACTIVATION COMPRESSION WITH K-MEANS METACLUSTERING

We will now outline our method for clustering the weights $M_\theta(\mathbf{z}_i)$ using K-means. Unlike Equation 4, the weights $M_\theta(\mathbf{z}_i)$ lie in $\mathbb{R}^{|\mathbf{w}|}$ rather than $\mathbb{R}$ as with the scalar weights $w_i$. Accordingly, each centroid $\mathbf{c}_i$ also resides in $\mathbb{R}^{|\mathbf{w}|}$. Therefore, our K-means objective must be adapted as follows:

$$\arg\min_{\mathcal{C}} \sum_{i=1}^{k} \sum_{M_\theta(\mathbf{z}) \in C_i} \left\| M_\theta(\mathbf{z}) - \mathbf{c}_i \right\|_2^2, \tag{7}$$

where $\mathbf{c}_i \in \mathbb{R}^{|\mathbf{w}|}$ denotes the centroid of cluster $C_i$.

The centroids are updated iteratively according to the standard K-means update rule:

$$\mathbf{c}_i = \frac{1}{|C_i|} \sum_{M_\theta(\mathbf{z}_j) \in C_i} M_\theta(\mathbf{z}_j), \tag{8}$$

ensuring that each centroid represents the mean of the weight vectors assigned to its cluster.

To record assignments, we define an index mapping vector $I \in \{1, 2, \ldots, k\}^N$, where the $j$-th entry indicates the cluster index of $M_\theta(\mathbf{z}_j)$. Formally,

$$I_j = \arg\min_{i \in \{1, \ldots, k\}} \left\| M_\theta(\mathbf{z}_j) - \mathbf{c}_i \right\|_2^2, \tag{9}$$

so that the mapping is expressed as $M_\theta(\mathbf{z}_j) \mapsto \mathbf{c}_{I_j}$.

### 3.3 Accuracy Recovery with Brief Fine-tuning

Once we have determined the mapping $I$ and the centroids $C$, we can remove the meta-learner $M_\theta$ and the embeddings $\mathbf{z}$ from the network as they are no longer needed (saving more memory in addition to compression). Since the clustered weights are an approximation of the original weights, we can recover any lost performance by fine-tuning the network. The fine-tuning process consists of an identical procedure to the original training, in which we optimized the centroid codebook for the task-specific loss for $\beta$ epochs, where $\beta << \alpha$ (the number of training epochs).

### 3.4 Storage Analysis

The compression factor relative to KAN is given by:

$$r = \frac{n|\mathbf{w}|b}{n\log_2(k) + |\mathbf{w}|kb} = \frac{nb}{n\log_2(k)/|\mathbf{w}| + kb} \tag{10}$$

As in Equation 5, $n$ denotes the number of connections, $k$ the number of centroids, and $b$ the number of bits used to represent each edge. Notably, the term $|\mathbf{w}|$ in the denominator reduces the relative contribution of $n\log_2(k)$ to the overall storage. Intuitively, this happens because each centroid stores more weight information ($|\mathbf{w}|$ entries per centroid), so the overhead of indexing and storing the centroids becomes proportionally smaller. As a result, KAN benefits more from the compression scheme than a standard MLP.

## 4 Experiments

We provide extensive experiments to showcase the efficacy of MetaCluster. In Section 4.1, we report the main results of the fully-connected and convolutional architecture on MNIST, CIFAR10, and CIFAR100 (Deng, 2012; Krizhevsky, 2009). Next, in Section 4.2, we provide the ablation study for our approach, covering the influence of the KAN meta-learner embedding size, basis coefficient vector sizes, and the cluster count on CIFAR10 (Krizhevsky, 2009). In Section 4.3 we report the computational cost incurred from MetaCluster. Finally, in Section 4.4, we provide high-dimensional equation modeling results with MetaCluster.

### 4.1 Image Classification Results

For all image classification experiments, we did a 90/10 training/validation split of the training data. We apply data augmentation to the train set for the convolutional architecture, following an identical procedure to Drokin (2024). We compare our approach across 24 model schemes. The base model for each scheme consists of KAN (B-spline basis) (Liu et al., 2024b), FastKAN (RBF basis) (Li, 2024), and GramKAN (Gram polynomial basis)(Drokin, 2024). In our naming conventions, we include *Cluster* to designate a clustered model, *Meta* to designate a model using a meta-learner, a design that originates from Zhao et al. (2025), and *Conv* to designate a model with a convolutional architecture (Bodner et al., 2024; Drokin, 2024). The fully-connected and convolutional models were clustered using 16 and 256 clusters, respectively. Each result is reported as an average of 5 runs.

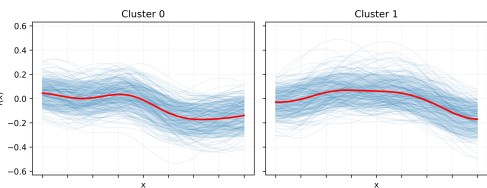

(a) The sample clusters for the first-layer of fully-connected FastKAN.

(b) The sample clusters for the first-layer of fully-connected MetaFastKAN.

Figure 2: A set of sample clusters showcasing the edge functions for each cluster (blue) and the centroid functions (red) for the first layer of the fully-connected (a) FastKAN and (b) MetaFastKAN models on Cifar-10.

Table 1: Classification accuracy and memory (KB) comparison of a fully-connected network.

| Model | MNIST | | CIFAR-10 | | CIFAR-100 | |
|---|---|---|---|---|---|---|
| | Memory | Acc. | Memory | Acc. | Memory | Acc. |
| KAN | 1,031.44 | $95.37 \pm 0.11$ | 3,064.95 | $47.52 \pm 0.16$ | 3,177.45 | $17.80 \pm 0.10$ |
| MetaKAN | 141.32 | $96.36 \pm 0.09$ | 410.82 | $46.28 \pm 0.39$ | 422.08 | $20.17 \pm 0.24$ |
| ClusterKAN | 13.84 | $61.37 \pm 2.77$ | 38.34 | $27.92 \pm 2.88$ | 39.75 | $6.20 \pm 0.31$ |
| + Fine-tune | 13.84 | $92.66 \pm 0.08$ | 38.34 | $43.71 \pm 0.27$ | 39.75 | $11.04 \pm 0.64$ |
| MetaClusterKAN | 13.84 | $96.06 \pm 0.10$ | 38.34 | $45.47 \pm 0.44$ | 39.75 | $18.97 \pm 0.15$ |
| + Fine-tune | 13.84 | $96.41 \pm 0.10$ | 38.34 | $45.95 \pm 0.33$ | 39.75 | $19.16 \pm 0.17$ |
| GPTVQ | 13.84 | $56.29 \pm 2.75$ | 38.34 | $24.27 \pm 1.59$ | 39.75 | $6.07 \pm 0.25$ |
| + Fine-tune | 13.84 | $93.88 \pm 0.25$ | 38.34 | $38.89 \pm 0.53$ | 39.75 | $8.96 \pm 0.18$ |
| FastKAN | 900.56 | $96.78 \pm 0.08$ | 2,676.83 | $49.10 \pm 0.29$ | 2,778.08 | $20.56 \pm 0.09$ |
| MetaFastKAN | 108.09 | $95.51 \pm 0.12$ | 316.35 | $47.40 \pm 0.46$ | 327.60 | $19.14 \pm 0.40$ |
| ClusterFastKAN | 20.54 | $32.34 \pm 4.16$ | 57.30 | $24.35 \pm 0.83$ | 58.71 | $6.72 \pm 0.41$ |
| + Fine-tune | 20.54 | $86.24 \pm 1.08$ | 57.30 | $38.89 \pm 0.35$ | 58.71 | $11.35 \pm 0.34$ |
| MetaClusterFastKAN | 20.54 | $94.87 \pm 0.29$ | 57.30 | $46.96 \pm 0.44$ | 58.71 | $18.66 \pm 0.48$ |
| + Fine-tune | 20.54 | $95.59 \pm 0.18$ | 57.30 | $47.60 \pm 0.37$ | 58.71 | $19.05 \pm 0.47$ |
| GPTVQ | 20.54 | $57.62 \pm 1.39$ | 57.30 | $20.84 \pm 0.71$ | 58.71 | $4.34 \pm 0.15$ |
| + Fine-tune | 20.54 | $92.98 \pm 0.35$ | 57.30 | $39.12 \pm 0.57$ | 58.71 | $9.24 \pm 0.37$ |
| GramKAN | 497.46 | $96.76 \pm 0.08$ | 1,477.48 | $49.57 \pm 0.43$ | 1,534.43 | $21.89 \pm 0.09$ |
| MetaGramKAN | 101.49 | $95.48 \pm 0.12$ | 297.49 | $48.24 \pm 0.19$ | 309.45 | $19.11 \pm 0.24$ |
| ClusterGramKAN | 13.96 | $68.72 \pm 1.65$ | 38.46 | $31.13 \pm 0.64$ | 40.58 | $6.34 \pm 0.43$ |
| + Fine-tune | 13.96 | $95.40 \pm 0.15$ | 38.46 | $45.81 \pm 0.66$ | 40.58 | $13.98 \pm 0.44$ |
| MetaClusterGramKAN | 14.15 | $93.80 \pm 0.72$ | 38.65 | $47.66 \pm 0.25$ | 40.76 | $18.18 \pm 0.20$ |
| + Fine-tune | 14.15 | $95.32 \pm 0.12$ | 38.65 | $48.47 \pm 0.30$ | 40.76 | $18.96 \pm 0.25$ |
| GPTVQ | 13.96 | $68.35 \pm 2.44$ | 38.46 | $16.85 \pm 1.63$ | 40.58 | $2.36 \pm 0.28$ |
| + Fine-tune | 13.96 | $91.29 \pm 0.47$ | 38.46 | $29.47 \pm 1.69$ | 40.58 | $6.59 \pm 0.61$ |

We also provide standard error bars. For our image classification experiments, a full description of our hyperparameters is given in Appendices A.1.1 and A.1.2.

The main results are reported in Tables 1 and 2. The general observation is that Meta variants have limited memory reduction but achieve classification accuracy comparable to their non-Meta counterparts, whereas clustering alone yields much higher memory compression, but at the cost of large accuracy drops. Our MetaCluster models retain the high accuracy of the Meta variants while achieving the memory compression of clustering.

### 4.1.1 Fully-Connected Network Image Classification

Table 1 demonstrates the high classification accuracy and compression factor of fully-connected MetaCluster models across all datasets and basis functions. For example, in the case of Meta-ClusterKAN, we can see that on Cifar-10, it achieves a compression factor of up to $10.7\times$ (from 410.82KB to 38.34KB) compared to the MetaKAN, and up to $79.9\times$ (from 3064.95 KB to 38.34KB) compared to the KAN. Furthermore, it achieves this compression factor while matching the classification accuracy of MetaKAN. Although the Meta variants are capable of achieving high accuracy without fine-tuning, this is not the case for the non-Meta variants. For example, in the case of GramKAN, it experiences a decrease in classification accuracy of up to $3.45\times$ (from 21.89% to 6.34%), which, when recovered with fine-tuning, remains a $1.57\times$ decrease in accuracy (from 21.89% to 13.98 %). Such results showcase the importance of leveraging a meta-learner to learn a set of activations that sit on a lower-dimensional subspace to aid in classification accuracy after clustering in fully-connected models. Table 1 also compares MetaCluster models against GPTVQ (Van Baalen et al., 2024), a popular vector quantization technique that performs hessian-aware weight sharing. From our experiments, we find that hessian-aware clustering on the KAN does not yield benefits and in some cases hurts downstream performance.

We further verify the importance of MetaCluster qualitatively by plotting edge functions and the respective centroids associated with the first layer of FastKAN and MetaFastKAN clusters in Figure 2. We provide the complete set of 16 clusters for each layer in Appendix A.5. From observing the functions associated with FastKAN and MetaFastKAN, it is apparent that those of MetaFastKAN are much closer to their respective centroids than those of FastKAN. Such a result once again demonstrates the importance of using meta-learners to find a lower-dimensional functional space before clustering.

Table 2: Classification accuracy and memory (KB) comparison of a convolutional network.

| MODEL | MNIST | | CIFAR-10 | | CIFAR-100 | |
|---|---|---|---|---|---|---|
| | MEMORY | ACC. | MEMORY | ACC. | MEMORY | ACC. |
| KANCONV | 13,634.00 | 99.46 ± 0.02 | 13,654.27 | 70.66 ± 2.89 | 13,744.62 | 23.99 ± 0.98 |
| METAKANCONV | 3,048.40 | 99.32 ± 0.05 | 3,052.90 | 68.36 ± 2.84 | 3,143.25 | 35.60 ± 1.78 |
| CLUSTERKANCONV | 429.95 | 92.16 ± 1.24 | 430.51 | 33.26 ± 1.30 | 520.87 | 10.81 ± 0.54 |
| + FINE-TUNE | 429.95 | 98.89 ± 0.09 | 430.51 | 61.80 ± 0.34 | 520.87 | 22.65 ± 0.81 |
| METACLUSTERKANCONV | 434.77 | 99.29 ± 0.06 | 435.34 | 62.48 ± 2.43 | 525.70 | 29.62 ± 1.65 |
| + FINE-TUNE | 434.77 | 99.35 ± 0.05 | 435.34 | 66.85 ± 3.11 | 525.70 | 35.11 ± 1.79 |
| FASTKANCONV | 13,632.09 | 99.11 ± 0.09 | 13,652.37 | 76.30 ± 0.11 | 13,742.72 | 46.36 ± 0.38 |
| METAFASTKANCONV | 3,041.57 | 99.00 ± 0.11 | 3,046.08 | 69.06 ± 1.15 | 3,136.43 | 39.60 ± 1.13 |
| CLUSTERFASTKANCONV | 428.03 | 62.61 ± 2.70 | 428.61 | 26.01 ± 3.55 | 518.97 | 5.16 ± 0.31 |
| + FINE-TUNE | 428.03 | 96.10 ± 0.09 | 428.61 | 57.45 ± 1.53 | 518.97 | 20.15 ± 0.70 |
| METACLUSTERFASTKANCONV | 427.97 | 99.01 ± 0.11 | 428.55 | 68.72 ± 1.06 | 518.91 | 38.75 ± 1.04 |
| + FINE-TUNE | 427.97 | 99.08 ± 0.06 | 428.55 | 69.10 ± 1.10 | 518.91 | 39.26 ± 1.10 |
| GRAMKANCONV | 7,586.47 | 99.40 ± 0.06 | 7,597.72 | 78.56 ± 1.14 | 7,688.07 | 48.25 ± 0.59 |
| METAGRAMKANCONV | 3,047.93 | 99.44 ± 0.05 | 3,052.43 | 80.91 ± 0.41 | 3,142.79 | 52.07 ± 0.63 |
| CLUSTERGRAMKANCONV | 418.93 | 61.17 ± 9.40 | 419.49 | 12.08 ± 0.66 | 509.85 | 1.77 ± 0.29 |
| + FINE-TUNE | 418.93 | 99.27 ± 0.08 | 419.49 | 75.74 ± 0.63 | 509.85 | 44.01 ± 0.42 |
| METACLUSTERGRAMKANCONV | 418.81 | 99.41 ± 0.06 | 419.38 | 79.25 ± 0.59 | 509.73 | 49.70 ± 0.51 |
| + FINE-TUNE | 418.81 | 99.47 ± 0.02 | 419.38 | 80.85 ± 0.41 | 509.73 | 51.20 ± 0.67 |

Table 3: Impact of basis coefficient count on memory (KB) footprint and classification accuracy.

| | METAFASTKAN | | METACLUSTERFASTKAN | | | METAFASTKANCONV | | METACLUSTERFASTKANCONV | | |
|---|---|---|---|---|---|---|---|---|---|---|
| COEFF | MEMORY | ACC% | MEMORY | ACC% | %CHG | MEMORY | ACC% | MEMORY | ACC% | %CHG |
| 5 | 316.35 | 47.40 ± 0.46 | 57.30 | 47.60 ± 0.37 | 0.42 | 3,046.08 | 69.97 ± 0.62 | 428.55 | 69.73 ± 0.63 | -0.34 |
| 8 | 316.76 | 46.29 ± 0.19 | 57.68 | 46.30 ± 0.24 | 0.02 | 3,046.52 | 68.89 ± 0.63 | 440.61 | 68.92 ± 0.60 | 0.05 |
| 10 | 317.03 | 46.63 ± 0.23 | 57.93 | 47.08 ± 0.27 | 0.95 | 3,046.81 | 68.23 ± 0.30 | 448.64 | 68.22 ± 0.27 | -0.01 |
| 15 | 317.71 | 45.26 ± 0.52 | 58.55 | 45.54 ± 0.44 | 0.62 | 3,047.54 | 64.87 ± 0.63 | 468.72 | 64.67 ± 0.55 | -0.30 |
| 20 | 318.40 | 46.11 ± 0.20 | 59.18 | 46.03 ± 0.27 | -0.17 | 3,048.26 | 61.16 ± 0.72 | 488.80 | 60.90 ± 0.75 | -0.43 |

### 4.1.2 CONVOLUTIONAL NETWORK IMAGE CLASSIFICATION

Table 2 demonstrates that, aside from MetaFastKANConv, the Meta variants match or exceed the classification accuracy of all the non-Meta variants at a reduced memory cost. For example, in the case of MetaKANConv on Cifar-10, the reduced memory cost is up to $4.5\times$ (from 13.7 MB to 3.1 MB) compared to KANConv. Upon clustering, we find that the compression factor increases by a factor of $7.1\times$ (from 3.1 MB to 435.34 KB). In total, compared to the KANConv, the Meta-ClusterKANConv achieves a reduction of up to $31.7\times$ (from 13.7 MB to 435.34 KB). For such a reduction in storage cost, after fine-tuning, we find there is a negligible impact on the downstream classification accuracy when comparing KANConv to MetaKANConv. Although the non-Meta variants recover accuracy from the fine-tuning step, the recovery still leaves them far behind their original state. For instance, upon clustering, FastKANConv experiences up to a $2.93\times$ decrease (from 76.30% to 26.01 % ) in classification accuracy. Then, when recovered with fine-tuning, it still retains a $1.32\times$ decrease (from 76.30% to 57.45 %) in classification accuracy. Once again, as with the fully-connected architecture, these results demonstrate the importance of our MetaCluster framework for maintaining accuracy with a high compression factor.

## 4.2 ABLATION STUDY

### 4.2.1 BASIS COEFFICIENT COUNT

We investigate the impact of the basis coefficient count (i.e., the number of radial basis functions) on downstream clustering performance. Since our technique employs a meta-learner to learn a lower-dimensional subspace, our approach should be resilient to changes in the basis coefficients we cluster. From Table 3, we can see that the initial accuracy of MetaFastKAN and MetaFastKANConv accuracy peaks at a coefficient count of 5 before dropping. However, even with these variations in accuracy for the coefficient count, the relative percentage change remains the same. This indicates that, regardless of the coefficient count, our MetaCluster framework remains capable of learning a lower-dimensional manifold, which facilitates easier clustering of weights.

### 4.2.2 CLUSTER COUNT

We verify that the MetaCluster framework scales well with the cluster count by comparing its cluster scaling properties with those of a non-meta KAN. As shown in Figure 3b, MetaCluster maintains accuracy well as clusters are reduced, especially after fine-tuning. For the convolutional model, decreasing the cluster count from 256 to 32 lowers MetaClusterFastKANConv accuracy by only 2%, whereas ClusterFastKANConv drops by 21.39% over the same range. Such results demonstrate that our approach has the capability of achieving even greater levels of compression, albeit at the expense of a minor accuracy impact.

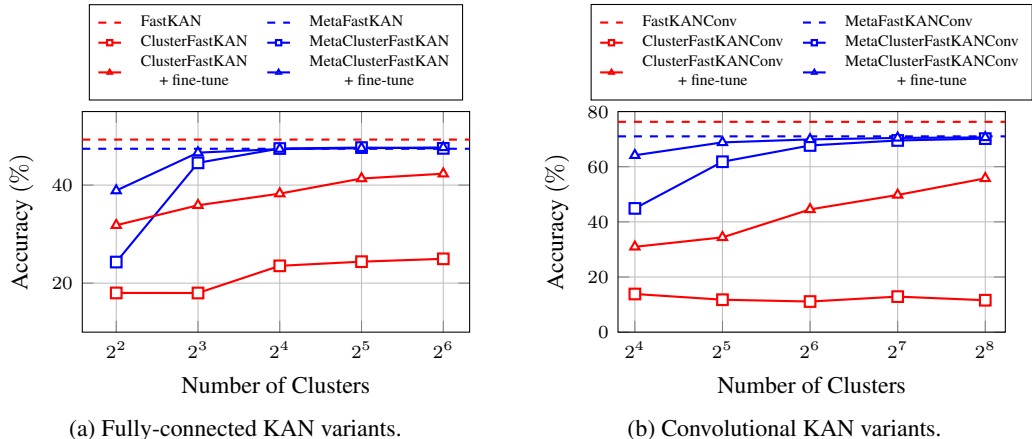

(a) Fully-connected KAN variants.  (b) Convolutional KAN variants.

Figure 3: Classification accuracy vs. number of clusters for FastKAN model variants.

### 4.2.3 META-LEARNER EMBEDDING SIZE

The embedding size of the meta-learner directly influences the ease of clustering the meta-learner generated weights. The general trend is that as the embedding dimension increases, it becomes increasingly difficult to cluster the generated weights appropriately. While we have qualitatively demonstrated this in Figure 1, where we see that increasing embedding dimension causes the plot to become less structured, we provide a quantitative analysis of the effect in Appendix A.2.2.

### 4.3 COMPUTATIONAL COST RESULTS

Table 4: Clustering times (in seconds)

| MODEL | MNIST | CIFAR-10 | CIFAR-100 |
|---|---|---|---|
| KAN | $0.029 \pm 0.011$ | $0.059 \pm 0.007$ | $0.080 \pm 0.022$ |
| METAKAN | $0.024 \pm 0.012$ | $0.048 \pm 0.011$ | $0.062 \pm 0.011$ |
| KANCONV | $3.587 \pm 0.145$ | $3.650 \pm 0.098$ | $3.542 \pm 0.114$ |
| METAKANCONV | $2.806 \pm 0.072$ | $2.812 \pm 0.067$ | $2.791 \pm 0.092$ |

As our approach is a three-stage framework consisting of a training stage, a clustering stage, and a fine-tuning stage, each stage will contribute to the computational cost during training. Furthermore, since our approach stores the KAN architecture as a codebook and indices that index the codebook, the forward cost of inference also incurs computational cost. We quantify the clustering stage cost in Table 4 and training/fine-tuning and inference costs in Table 5. Each metric is reported in seconds, averaged over 10 runs, with standard error, on an RTX 5090 GPU. The training and inference times are reported as the time to complete one epoch and the time to evaluate on the test set, respectively.

From observing Table 4 we can see that compared to training times the clustering times are negligible (e.g., 0.029 s once vs 2.155 s per epoch). We can also see that the time required to train MetaKAN per epoch is slightly longer than that for KAN. Similarly, during fine-tuning (which runs for much fewer epochs than the training stage), the time per epoch is slightly longer for MetaClusterKAN (an identical architecture to ClusterKAN) than for KAN. Such a trend deviates in inference, where the MetaClusterKAN is as fast as KAN. Similar trends are seen with the convolutional versions of the models. Such results show there is minimal overhead from MetaCluster.

Table 5: Training/fine-tuning and inference times (in seconds)

| Model | MNIST | | CIFAR-10 | | CIFAR-100 | |
|---|---|---|---|---|---|---|
| | Train | Inference | Train | Inference | Train | Inference |
| KAN | $2.155 \pm 0.047$ | $0.320 \pm 0.005$ | $2.661 \pm 0.049$ | $0.510 \pm 0.008$ | $2.730 \pm 0.060$ | $0.509 \pm 0.007$ |
| MetaKAN | $2.501 \pm 0.055$ | $0.337 \pm 0.003$ | $2.963 \pm 0.056$ | $0.538 \pm 0.012$ | $2.956 \pm 0.096$ | $0.526 \pm 0.008$ |
| MetaClusterKAN | $2.284 \pm 0.048$ | $0.323 \pm 0.002$ | $3.259 \pm 0.045$ | $0.521 \pm 0.008$ | $3.252 \pm 0.091$ | $0.542 \pm 0.019$ |
| KANConv | $9.158 \pm 0.184$ | $0.922 \pm 0.009$ | $10.105 \pm 0.085$ | $1.130 \pm 0.003$ | $10.370 \pm 0.162$ | $1.204 \pm 0.023$ |
| MetaKANConv | $10.039 \pm 0.228$ | $0.930 \pm 0.009$ | $11.024 \pm 0.080$ | $1.184 \pm 0.025$ | $10.946 \pm 0.097$ | $1.223 \pm 0.011$ |
| MetaClusterKANConv | $9.931 \pm 0.160$ | $0.951 \pm 0.017$ | $10.605 \pm 0.144$ | $1.149 \pm 0.032$ | $10.929 \pm 0.156$ | $1.202 \pm 0.004$ |

## 4.4 Equation Modeling Results

Our high-dimensional equation modeling results explored the ability of MetaCluster to model 1000-dimensional equations bounded on $[-1, 1]$. All KAN models used were identical to our fully-connected image classification experiments. We explored the equations $f_1(x) = \exp\left(\frac{1}{n}\sum \sin^2\left(\frac{\pi x}{2}\right)\right)$, $f_2(x) = \sum x^2 + x^3$, $f_3(x) = \exp\left(-\frac{1}{n}\sum x^2\right)$ following a similar procedure to Zhao et al. (2025). The clustered models used a cluster size of 4. Each reported MSE result is an average over 10 runs. We provide the complete set of hyperparameters in Appendix A.1.3.

Table 6: MSE and memory (KB) comparison in high-dimensional equation modeling.

| Model | $f_1(x) = \exp\left(\frac{1}{n}\sum \sin^2\left(\frac{\pi x}{2}\right)\right)$ | | $f_2(x) = \sum x^2 + x^3$ | | $f_3(x) = \exp\left(-\frac{1}{n}\sum x^2\right)$ | |
|---|---|---|---|---|---|---|
| | Memory | MSE | Memory | MSE | Memory | MSE |
| KAN | 1303.6 | $5.061 \times 10^{-5}$ | 1303.6 | $2.698 \times 10^{-1}$ | 1303.6 | $5.297 \times 10^{-7}$ |
| MetaKAN | 181.6 | $3.048 \times 10^{-5}$ | 181.6 | $2.407 \times 10^{0}$ | 181.6 | $2.762 \times 10^{-6}$ |
| ClusterKAN | 10.5 | $9.847 \times 10^{-1}$ | 10.5 | $1.656 \times 10^{2}$ | 10.5 | $1.266 \times 10^{-1}$ |
| + fine-tune | 10.5 | $2.239 \times 10^{-4}$ | 10.5 | $8.924 \times 10^{-1}$ | 10.5 | $7.816 \times 10^{-6}$ |
| MetaClusterKAN | 10.5 | $9.027 \times 10^{-4}$ | 10.5 | $1.584 \times 10^{0}$ | 10.5 | $3.750 \times 10^{-4}$ |
| + fine-tune | 10.5 | $2.697 \times 10^{-5}$ | 10.5 | $5.039 \times 10^{-1}$ | 10.5 | $7.547 \times 10^{-7}$ |

From Table 6, we can see that MetaClusterKAN achieves identical or, in some cases, reduced mean squared error (MSE) than the original KAN model after clustering. MetaClusterKAN can achieve these equation modeling capabilities with a 124.1x reduction in memory. Such a result is not the case for ClusterKAN, which shows a significant increase in MSE after clustering (from $5.061 \times 10^{-5}$ to $2.239 \times 10^{-4}$). These results demonstrate that the principles from MetaCluster extend beyond image classification into the equation modeling domain.

# 5 Related Works

## 5.1 Kolmogorov-Arnold Networks

KANs have evolved rapidly since their introduction, with numerous variants exploring different basis functions such as B-splines, radial basis functions (RBFs), Chebyshev and Legendre polynomials, Gram polynomials, wavelets, and rational functions (Liu et al., 2024b; Li, 2024; SS et al., 2024; Bozorgasl & Chen, 2024; Aghaei, 2024). These architectures have demonstrated strong performance in scientific computing and equation modeling (Li et al., 2025; Wang et al., 2025; Coffman & Chen, 2025; Koenig et al., 2024) and have recently extended to computer vision through designs such as the Kolmogorov-Arnold Transformer (Yang & Wang, 2024; Raffel & Chen, 2025). Despite these successes, KANs face persistent challenges, including training instability, computational overhead, and a substantial increase in parameter count compared to MLPs (Yu et al., 2024; Chen et al., 2024). Our work directly targets the memory scaling obstacle of KANs, creating topologically identical KANs that are smaller than comparable MLPs while maintaining accuracy.

## 5.2 HyperNetworks

Hypernetworks reduce trainable parameter counts by replacing task or instance-specific weights with a shared generator that predicts them on demand. The idea originates from early meta-learning

works such as Ba et al. (2016), where a small network generates classifier weights from context, and Bertinetto et al. (2016), which learns to emit a one-shot tracker's parameters conditioned on a single exemplar. Ha et al. (2016) then formalized Hypernetworks for CNNs and RNNs, showing that a compact hypernetwork can match the accuracy of a standard model while drastically cutting the number of directly optimized parameters. Later extensions use task embeddings to condition a single hypernetwork for multi-task or multi-objective weight generation (Savarese & Maire, 2019; Navon et al., 2020) .

MetaKANs (Zhao et al., 2025) bring this paradigm to KANs by observing that the dominant cost in a KAN is storing the coefficients of every univariate activation. Instead of optimizing all coefficients directly, a small meta-learner maps per-activation embeddings to basis coefficients, capturing a shared rule for weight generation across activations (Zhao et al., 2025). In contrast, our method uses the meta-learner only during training to impose a clusterable geometry on per-edge coefficient embeddings. Then, at inference, we dispense with any hypernetwork, incurring zero runtime overhead while achieving even greater parameter efficiency.

## 5.3 WEIGHT SHARING

Weight sharing compresses networks by restricting each layer's parameters to a small set of shared centroids and storing a codebook, along with per-weight indices. Scalar quantization approaches, popularized by Han et al. (2015) introduced a pipeline combining of magnitude pruning and K-means weight sharing. Subsequent theory linked quantization error to loss curvature and leveraged a Hessian-weighted K-means to cluster (Choi et al., 2016). Differentiable K-means (DKM) extended this idea by jointly optimizing centroids and assignments with task loss (Cho et al., 2021).

Beyond scalar quantization, vector quantization offers a natural extension by clustering weight vectors rather than individual scalars, which early work applied to improve compression–accuracy trade-offs (Gong et al., 2014). More recently, the rise of large language models has driven specialized vector quantization approaches, which leverage Hessian information and the inherent sparsity of LLMs to optimize vector-level assignments (Liu et al., 2024a; Egiazarian et al., 2024; Van Baalen et al., 2024; Tseng et al., 2024). We demonstrate in Section 4.1.1 that such Hessian information is not effective for KANs, a result that is in line with Hessian-aware quantization techniques reported in Fuad & Chen (2025). Building on vector quantization research, our MetaCluster framework is the first to apply weight-sharing principles to the KAN successfully.

It is worth noting that, while weight-sharing is one avenue for model compression, bit-width quantization is an alternative avenue, which is largely orthogonal and complementary to the proposed approach. Applying quantization on top of MetaCluster (e.g., quantizing the codebook of centroids) has the potential to achieve a further reduction in the memory footprint.

## 6 CONCLUSION

We introduced MetaCluster, a compression framework that makes KANs practical at scale by introducing the novel combination of meta-learned manifold shaping with weight sharing. A lightweight meta-learner maps low-dimensional embeddings to per-edge basis coefficients, constraining KAN activations to lie on a compact manifold that is amenable to clustering. We then apply K-means in coefficient space, replace per-edge parameters with codebook centroids and compact indices, discard the meta-learner, and briefly fine-tune centroids to recover any loss. This design directly targets the dimensionality driver of KAN memory, yielding a storage advantage that grows with the number of coefficients per edge.

Across standard KANs and ConvKANs on MNIST, CIFAR-10, and CIFAR-100, MetaCluster achieves up to $80\times$ reduction in parameter storage without degrading accuracy. Furthermore, on equation modeling it achieves an even more impressive $124.1\times$ reduction in parameter storage with no drop in performance. Visualizations and ablations demonstrate that manifold shaping is crucial for high-quality clustering in high dimensions, and that KANs benefit particularly from weight sharing compared to MLPs, as each centroid amortizes many coefficients.

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

# A APPENDIX

## A.1 HYPERPARAMETERS

### A.1.1 IMAGE CLASSIFICATION FULLY-CONNECTED SETUP

Our fully-connected architecture follows Zhao et al. (2025) in that we stack two fully-connected KAN layers, with a 32-dimensional hidden state between them. Each KAN variant uses a SiLU activation (Elfwing et al., 2018). The meta-learner variants contained a meta-learner with a hidden dimension of 32. Concretely:

- **KAN:** degree–3 B-splines, grid range $[-1, 1]$, grid size 5 (Liu et al., 2024b).
- **FastKAN:** 8 RBFs over $[-2, 2]$ (Li, 2024).
- **GramKAN:** degree–3 polynomial (Drokin, 2024).

We train with AdamW (Loshchilov & Hutter, 2017) and for at most 50 epochs, with early stopping (patience = 10). After the final epoch, we cluster the learned weights into 16 groups, then fine-tune for an additional 5 epochs.For our fully-connected architecture experiments Table 7 reports the full set of hyperparameters for our KAN variants and Table 8 reports the full set of hyperparameters for our MetaKAN variants.

Table 7: Hyperparameters for fully-connected KAN variants.

| HYPERPARAMETER | KAN | FASTKAN | GRAMKAN |
|---|---|---|---|
| HIDDEN DIMENSION | 32 | 32 | 32 |
| ACTIVATION | SILU | SILU | SILU |
| DEGREE | 3 (SPLINE) | - | 3 |
| GRID RANGE | $[-1, 1]$ | $[-2, 2]$ | - |
| GRID SIZE | 5 | 8(NUM GRIDS) | - |
| OPTIMIZER | ADAMW | ADAMW | ADAMW |
| LEARNING RATE | $1 \times 10^{-4}$ | $1 \times 10^{-3}$ | $1 \times 10^{-3}$ |
| LEARNING RATE FOR FINETUNING (LR_C) | $1 \times 10^{-4}$ | $1 \times 10^{-4}$ | $1 \times 10^{-4}$ |
| BATCH SIZE | 128 | 128 | 128 |
| EPOCHS | 50 | 50 | 50 |
| EARLY STOPPING PATIENCE | 10 | 10 | 10 |
| CLUSTERED TRAINING EPOCHS | 5 | 5 | 5 |
| EARLY STOPPING PATIENCE (FINE-TUNING) | 3 | 3 | 3 |
| NUMBER OF CLUSTERS | 16 | 16 | 16 |

Table 8: Hyperparameters for fully-connected MetaKAN variants.

| HYPERPARAMETER | METAKAN | METAFASTKAN | METAGRAMKAN |
|---|---|---|---|
| HIDDEN DIMENSION | 32 | 32 | 32 |
| EMBEDDING DIMENSION | 1 | 1 | 1 |
| ACTIVATION | SILU | SILU | SILU |
| DEGREE | 3 (SPLINE) | - | 3 |
| GRID RANGE | $[-1, 1]$ | $[-2, 2]$ | - |
| GRID SIZE | 5 | 8(NUM-GRIDS) | - |
| OPTIMIZER SET | DOUBLE | DOUBLE | DOUBLE |
| OPTIMIZER | ADAMW | ADAMW | ADAMW |
| LEARNING RATE FOR META-LEARNER (LR_H) | $5 \times 10^{-4}$ | $1 \times 10^{-3}$ | $1 \times 10^{-4}$ |
| LEARNING RATE FOR EMBEDDINGS (LR_C) | $5 \times 10^{-3}$ | $1 \times 10^{-2}$ | $1 \times 10^{-3}$ |
| LEARNING RATE FOR FINETUNING (LR_C) | $1 \times 10^{-4}$ | $1 \times 10^{-4}$ | $1 \times 10^{-4}$ |
| BATCH SIZE | 128 | 128 | 128 |
| EPOCHS | 50 | 50 | 50 |
| CLUSTERED TRAINING EPOCHS | 5 | 5 | 5 |
| EARLY STOPPING PATIENCE | 10 | 10 | 10 |
| EARLY STOPPING PATIENCE (FINE-TUNING) | 3 | 3 | 3 |
| NUMBER OF CLUSTERS | 16 | 16 | 16 |

### A.1.2 IMAGE CLASSIFICATION CONVOLUTIONAL SETUP

Our convolutional architecture is taken from Drokin (2024): four convolutional layers with channel progression $[32, 64, 128, 512]$, each using $3 \times 3$ kernels, stride 1, and padding 1. As with the fully-connected experiments, each KAN variant uses a SiLU activation (Elfwing et al., 2018) and the

meta-learner versions use a meta-learner with a hidden dimension of 32. We replace the usual activation with our kernels:

- **KANConv:** degree–3 B-splines, grid range $[-3, 3]$, grid size 5 (Liu et al., 2024b).
- **FastKANConv:** 8 RBFs on $[-3, 3]$ (Li, 2024).
- **GramKANConv:** degree–3 polynomial (Drokin, 2024).

Training again uses AdamW for up to 150 epochs with early stopping (patience = 10). Final weights are clustered into 16 groups and fine-tuned for 5 epochs. For our convolutional architecture experiments Table 9 reports the full set of hyperparameters for our KAN variants and Table 10 reports the full set of hyperparameters for our MetaKAN variants.

Table 9: Hyperparameters for convolutional non-meta KAN variants.

| HYPERPARAMETER | KAN | FASTKAN | KAGN |
|---|---|---|---|
| HIDDEN DIMENSION | [32, 64, 128, 256] | [32, 64, 128, 256] | [32, 64, 128, 256] |
| ACTIVATION | SILU | SILU | SILU |
| DEGREE | 3 (SPLINE) | - | 3 |
| GRID RANGE | $[-3, 3]$ | $[-3, 3]$ | - |
| GRID SIZE | 5 | 8(NUM-GRIDS) | - |
| DROPOUT | 0.25 | 0.25 | 0.25 |
| DROPOUT (LINEAR LAYERS) | 0.5 | 0.5 | 0.5 |
| OPTIMIZER | ADAMW | ADAMW | ADAMW |
| LEARNING RATE FOR FINETUNING (LR_C) | $1 \times 10^{-5}$ | $1 \times 10^{-5}$ | $1 \times 10^{-5}$ |
| LEARNING RATE (LR) | $1 \times 10^{-3}$ | $1 \times 10^{-3}$ | $1 \times 10^{-3}$ |
| BATCH SIZE | 128 | 128 | 128 |
| EPOCHS | 150 | 150 | 150 |
| CLUSTERED TRAINING EPOCHS | 5 | 5 | 5 |
| EARLY STOPPING PATIENCE | 10 | 10 | 10 |
| EARLY STOPPING PATIENCE (FINE-TUNING) | 3 | 3 | 3 |
| NUMBER OF CLUSTERS | 256 | 256 | 256 |
| CONVOLUTION GROUPS | 1 | 1 | 1 |

Table 10: Hyperparameters for convolutional MetaKAN variants.

| HYPERPARAMETER | METAKAN | METAFASTKAN | METAKAGN |
|---|---|---|---|
| HIDDEN DIMENSION | [32, 64, 128, 256] | [32, 64, 128, 256] | [32, 64, 128, 256] |
| EMBEDDING DIMENSION | 2 | 2 | 1 |
| ACTIVATION | SILU | SILU | SILU |
| DEGREE | 3 (SPLINE) | - | 3 |
| GRID RANGE | $[-3, 3]$ | $[-3, 3]$ | - |
| GRID SIZE | 5 | 8(NUM-grid) | - |
| OPTIMIZER SET | DOUBLE | DOUBLE | DOUBLE |
| OPTIMIZER | ADAMW | ADAMW | ADAMW |
| LEARNING RATE FOR META-LEARNER (LR_H) | $1 \times 10^{-4}$ | $1 \times 10^{-4}$ | $1 \times 10^{-4}$ |
| LEARNING RATE EMBEDDING (LR_E) | $5 \times 10^{-3}$ | $5 \times 10^{-3}$ | $5 \times 10^{-3}$ |
| LEARNING RATE FOR FINETUNING (LR_C) | $1 \times 10^{-5}$ | $1 \times 10^{-5}$ | $1 \times 10^{-5}$ |
| GLOBAL LEARNING RATE (LR) | $1 \times 10^{-3}$ | $5 \times 10^{-3}$ | $5 \times 10^{-3}$ |
| BATCH SIZE | 128 | 128 | 128 |
| EPOCHS | 150 | 150 | 150 |
| CLUSTERED TRAINING EPOCHS | 5 | 5 | 5 |
| EARLY STOPPING PATIENCE | 10 | 10 | 10 |
| EARLY STOPPING PATIENCE (FINE-TUNING) | 3 | 3 | 3 |
| NUMBER OF CLUSTERS | 256 | 256 | 256 |
| EMBEDDING SCHEDULER | YES | YES | YES |
| HYPERNET SCHEDULER | YES | YES | YES |
| DROPOUT | 0.25 | 0.25 | 0.25 |
| DROPOUT (LINEAR LAYERS) | 0.5 | 0.5 | 0.5 |
| CONVOLUTION GROUPS | 1 | 1 | 1 |

### A.1.3 EQUATION MODELING SETUP

We report our complete set of hyperparameter for our equation modeling experiments in Table 11.

Table 11: Hyperparameters for fully-connected KAN equation modeling experiments.

| HYPERPARAMETER | KAN |
|---|---|
| HIDDEN DIMENSION | 32 |
| ACTIVATION | SiLU |
| DEGREE | 3 (SPLINE) |
| GRID RANGE | $[-1, 1]$ |
| GRID SIZE | 5 |
| OPTIMIZER | ADAMW |
| LEARNING RATE | $1 \times 10^{-4}$ |
| LEARNING RATE FOR FINETUNING (LR_C) | $1 \times 10^{-4}$ |
| BATCH SIZE | 256 |
| EPOCHS | 100 |
| CLUSTERED TRAINING EPOCHS | 10 |
| NUMBER OF CLUSTERS | 4 |
| DOMAIN OF DATAPOINTS | [-1,1] |
| TRAIN SET SAMPLES | 50,000 |
| TEST SET SAMPLES | 20,000 |

## A.2 ABLATIONS EXTENDED

We present this extended ablation study to provide a more detailed analysis of the impact of various hyperparameters on accuracy and the effectiveness of fine-tuning across different settings. These extended results provide further insight into how variations in embedding size and coefficient count, along with fine-tuning, influence performance differences in clustering across the different KAN model variants and network architectures.

### A.2.1 BASIS COEFFICIENT COUNT EXTENDED

Table 12 provides detailed insight into the influence of the fine-tuning stage of the MetaCluster framework on different basis coefficient counts (i.e. the grid size or number of radial basis functions). We can see that since all the original classification accuracy is retained after clustering, there are minimal additional accuracy gains offered by fine-tuning the clustered model.

Table 12: Detailed results of grid sizes versus accuracy and memory for both fully-connected and convolutional MetaFastKAN networks on Cifar-10.

| Model | Grid Size | Grid Range | Embed Dim | # Clusters | Model Accuracy | Clustered Accuracy | Fine-tuned Accuracy | Mem Before Clustering (KB) | Mem After Clustering (KB) |
|---|---|---|---|---|---|---|---|---|---|
| MetaClusterFastKAN | 5 | -2,2 | 1 | 16 | 47.41 | 47.46 | 47.35 | 316.35 | 57.30 |
| MetaClusterFastKAN | 8 | -2,2 | 1 | 16 | 47.62 | 46.96 | 47.80 | 316.76 | 57.68 |
| MetaClusterFastKAN | 10 | -2,2 | 1 | 16 | 46.91 | 47.04 | 47.49 | 317.03 | 57.93 |
| MetaClusterFastKAN | 15 | -2,2 | 1 | 16 | 45.88 | 45.15 | 45.74 | 317.71 | 58.55 |
| MetaClusterFastKAN | 20 | -2,2 | 1 | 16 | 43.57 | 43.75 | 44.31 | 318.40 | 59.18 |
| MetaClusterFastKANConv | 5 | -3,3 | 2 | 256 | 71.42 | 71.06 | 71.38 | 3,046.08 | 428.55 |
| MetaClusterFastKANConv | 8 | -3,3 | 2 | 256 | 69.61 | 68.66 | 69.24 | 3,046.52 | 440.61 |
| MetaClusterFastKANConv | 10 | -3,3 | 2 | 256 | 66.78 | 66.58 | 66.74 | 3,046.81 | 448.64 |
| MetaClusterFastKANConv | 15 | -3,3 | 2 | 256 | 64.24 | 63.99 | 64.33 | 3,047.54 | 468.72 |
| MetaClusterFastKANConv | 20 | -3,3 | 2 | 256 | 54.41 | 54.45 | 54.48 | 3,048.26 | 488.80 |

### A.2.2 META-LEARNER EMBEDDING SIZE EXTENDED

Our ablation exploring the influence of the embedding dimension aims to quantify the influence of the embedding size on the downstream clustering performance. We report these results in Table 13. From Table 13, we can see that in both the fully-connected and convolutional architectures, as the embedding dimension increases, the classification accuracy after clustering decreases. This validates our assumption, which we developed based on Figure 1, that finding a lower-dimensional subspace improves downstream clustering performance. During the fine-tuning stage, we can recover most of the accuracy decrease experienced by choosing a higher-dimensional embedding. Furthermore, the higher-dimensional choice of embedding does not affect the MetaCluster model memory footprint.

Table 13: Detailed results of embedding dimension versus accuracy for both fully-connected and convolutional MetaFastKAN networks on Cifar-10.

| Model | Num-grid | Grid Range | Embed Dim | # Clusters | Model Accuracy | Clustered Accuracy | Fine-tuned Accuracy | Mem Before Clustering (KB) | Mem After Clustering (KB) |
|---|---|---|---|---|---|---|---|---|---|
| MetaClusterFastKAN | 5 | -2,2 | 4 | 16 | 48.66 | 38.28 | 44.63 | 1,202.50 | 57.30 |
| MetaClusterFastKAN | 5 | -2,2 | 3 | 16 | 47.28 | 39.11 | 45.20 | 907.10 | 57.30 |
| MetaClusterFastKAN | 5 | -2,2 | 2 | 16 | 48.19 | 45.48 | 47.29 | 611.72 | 57.30 |
| MetaClusterFastKANConv | 5 | -3,3 | 4 | 256 | 71.16 | 66.46 | 69.58 | 6,077.09 | 428.55 |
| MetaClusterFastKANConv | 5 | -3,3 | 3 | 256 | 73.78 | 70.21 | 72.39 | 4,561.59 | 428.55 |
| MetaClusterFastKANConv | 5 | -3,3 | 2 | 256 | 71.42 | 71.05 | 71.33 | 3,046.08 | 428.55 |

## A.3 ALTERNATIVE DISTANCE METRIC

Our use of Euclidean distance follows directly from the properties of the K-means algorithm. K-means minimizes the sum of squared Euclidean distances between points and their assigned centroids, and its update step relies on this metric to compute cluster means in closed form. Substituting a different metric (e.g., cosine similarity or a learned distance) would break this property and require a fundamentally different clustering procedure, such as K-medoids or metric-learning-based clustering, which introduces additional complexity and computational overhead.

We qualitatively justify our use of Euclidean distance in Appendix A.5, where we can see with our current methodology that the functions of MetaFastKAN already lie near the centroids due to the reduced functional space, unlike FastKAN. These visualizations confirm that the manifold is sufficiently compact and well-aligned, making Euclidean distance not only effective but sufficient for capturing meaningful relationships.

Table 14: Classification accuracy and memory (KB) comparison of a fully-connected network with clustering using cosine-similarity.

| MODEL | MNIST | | CIFAR-10 | | CIFAR-100 | |
|---|---|---|---|---|---|---|
| | MEMORY | ACC. | MEMORY | ACC. | MEMORY | ACC. |
| KAN | 1,031.44 | 95.70 | 3,064.95 | 47.57 | 3,177.45 | 17.60 |
| METAKAN | 141.32 | 96.56 | 410.82 | 47.59 | 422.08 | 19.30 |
| CLUSTERKAN | 13.84 | 44.37 | 38.34 | 24.94 | 39.75 | 7.50 |
| + FINE-TUNE | 13.84 | 91.89 | 38.34 | 42.98 | 39.75 | 11.13 |
| METACLUSTERKAN | 13.84 | 87.42 | 38.34 | 40.73 | 39.75 | 15.74 |
| + FINE-TUNE | 13.84 | 94.55 | 38.34 | 43.60 | 39.75 | 16.78 |
| FASTKAN | 900.56 | 96.91 | 2,676.83 | 48.40 | 2,778.08 | 20.68 |
| METAFASTKAN | 108.09 | 95.62 | 316.35 | 46.14 | 327.60 | 18.15 |
| CLUSTERFASTKAN | 20.54 | 31.60 | 57.30 | 22.47 | 58.71 | 6.44 |
| + FINE-TUNE | 20.54 | 87.36 | 57.30 | 39.00 | 58.71 | 10.91 |
| METACLUSTERFASTKAN | 20.54 | 94.89 | 57.30 | 43.88 | 58.71 | 16.49 |
| + FINE-TUNE | 20.54 | 95.66 | 57.30 | 45.84 | 58.71 | 17.64 |
| GRAMKAN | 497.46 | 97.02 | 1,477.48 | 50.10 | 1,534.43 | 21.76 |
| METAGRAMKAN | 101.49 | 95.76 | 297.49 | 48.57 | 309.45 | 18.96 |
| CLUSTERGRAMKAN | 13.96 | 62.98 | 38.46 | 27.61 | 40.58 | 2.91 |
| + FINE-TUNE | 13.96 | 90.49 | 38.46 | 43.70 | 40.58 | 11.58 |
| METACLUSTERGRAMKAN | 14.15 | 92.45 | 38.65 | 42.68 | 40.76 | 17.26 |
| + FINE-TUNE | 14.15 | 94.61 | 38.65 | 49.05 | 40.76 | 18.59 |

For quantitative validation, we explored cosine similarity as an alternative metric. The results, shown in Table 14, indicate a slight decrease in downstream accuracy after compression compared to Euclidean distance. While cosine similarity emphasizes directional alignment, it ignores vector magnitudes, which appear to carry meaningful information in our setting. Euclidean distance, by contrast, leverages both direction and scale, making it better suited for our representation space. This advantage likely arises from the meta-learning stage, which organizes coefficient vectors into a structured manifold where absolute distances remain informative.

## A.4 IMAGE CLASSIFICATION WITH GAUSSIAN MIXTURE MODEL CLUSTERING

Our Gaussian mixture model clustering experiments followed the same setup described in Appendices A.1.1 and A.1.2 for the fully connected and convolutional models, with one key difference:

clustering was performed using GMMs rather than K-Means. The results for both architectures are reported in Tables 15 and 16, respectively.

Table 15: Classification accuracy and memory (KB) comparison of a fully-connected network using Gaussian mixture model clustering.

| MODEL | MNIST | | CIFAR-10 | | CIFAR-100 | |
|---|---|---|---|---|---|---|
| | MEMORY | ACC. | MEMORY | ACC. | MEMORY | ACC. |
| KAN | 1,031.44 | 95.70 | 3,064.95 | 47.57 | 3,177.45 | 17.60 |
| METAKAN | 141.32 | 96.56 | 410.82 | 47.59 | 422.08 | 19.30 |
| CLUSTERKAN | 13.84 | 65.66 | 38.34 | 22.83 | 39.75 | 4.77 |
| + FINE-TUNE | 13.84 | 90.47 | 38.34 | 42.63 | 39.75 | 11.37 |
| METACLUSTERKAN | 13.84 | 96.22 | 38.34 | 45.87 | 39.75 | 18.61 |
| + FINE-TUNE | 13.84 | 96.55 | 38.34 | 46.64 | 39.75 | 18.85 |
| FASTKAN | 900.56 | 96.91 | 2,676.83 | 48.40 | 2,778.08 | 20.68 |
| METAFASTKAN | 108.09 | 95.62 | 316.35 | 46.14 | 327.60 | 18.15 |
| CLUSTERFASTKAN | 20.54 | 51.68 | 57.30 | 21.36 | 58.71 | 6.16 |
| + FINE-TUNE | 20.54 | 89.03 | 57.30 | 36.26 | 58.71 | 10.46 |
| METACLUSTERFASTKAN | 20.54 | - | 57.30 | 45.5 | 58.71 | 16.78 |
| + FINE-TUNE | 20.54 | - | 57.30 | 46.57 | 58.71 | 17.64 |
| GRAMKAN | 497.46 | 97.02 | 1,477.48 | 50.10 | 1,534.43 | 21.76 |
| METAGRAMKAN | 101.49 | 95.76 | 297.49 | 48.57 | 309.45 | 18.96 |
| CLUSTERGRAMKAN | 13.96 | 66.58 | 38.46 | 29.84 | 40.58 | 4.61 |
| + FINE-TUNE | 13.96 | 94.39 | 38.46 | 44.44 | 40.58 | 10.20 |
| METACLUSTERGRAMKAN | 14.15 | 94.98 | 38.65 | 46.88 | 40.76 | 17.39 |
| + FINE-TUNE | 14.15 | 95.33 | 38.65 | 49.25 | 40.76 | 18.33 |

Tables 15 and 16 demonstrate that using Gaussian mixture models for clustering the functional edges of the KAN offers no benefit over K-Means. They also demonstrate that our MetaCluster three-stage framework is still needed to achieve high-quality functional clustering.

Table 16: Classification accuracy and memory (KB) comparison of a convolutional network using Gaussian mixture model clustering.

| MODEL | MNIST | | CIFAR-10 | | CIFAR-100 | |
|---|---|---|---|---|---|---|
| | MEMORY | ACC. | MEMORY | ACC. | MEMORY | ACC. |
| KANCONV | 13,634.00 | 99.45 | 13,654.27 | 74.05 | 13,744.62 | 26.00 |
| METAKANCONV | 3,048.40 | 99.49 | 3,052.90 | 68.23 | 3,143.25 | 36.23 |
| CLUSTERKANCONV | 429.95 | 88.08 | 430.51 | 24.22 | 520.87 | 4.40 |
| + FINE-TUNE | 429.95 | 98.53 | 430.51 | 52.62 | 520.87 | 18.74 |
| METACLUSTERKANCONV | 434.77 | 99.06 | 435.34 | 54.54 | 525.70 | 15.05 |
| + FINE-TUNE | 434.77 | 99.49 | 435.34 | 65.79 | 525.70 | 35.36 |
| FASTKANCONV | 13,632.09 | 99.41 | 13,652.37 | 76.48 | 13,742.72 | 47.10 |
| METAFASTKANCONV | 3,041.57 | 99.26 | 3,046.08 | 69.87 | 3,136.43 | 40.99 |
| CLUSTERFASTKANCONV | 428.03 | 46.40 | 428.61 | 12.84 | 518.97 | 2.68 |
| + FINE-TUNE | 428.03 | 85.63 | 428.61 | 43.03 | 518.97 | 5.52 |
| METACLUSTERFASTKANCONV | 427.97 | 98.64 | 428.55 | 67.54 | 518.91 | 37.18 |
| + FINE-TUNE | 427.97 | 99.17 | 428.55 | 69.44 | 518.91 | 40.22 |
| GRAMKANCONV | 7,586.47 | 99.42 | 7,597.72 | 75.19 | 7,688.07 | 47.23 |
| METAGRAMKANCONV | 3,047.93 | 99.46 | 3,052.43 | 81.52 | 3,142.79 | 53.27 |
| CLUSTERGRAMKANCONV | 418.93 | 10.28 | 419.49 | 10.04 | 509.85 | 1.00 |
| + FINE-TUNE | 418.93 | 99.27 | 419.49 | 72.73 | 509.85 | 25.00 |
| METACLUSTERGRAMKANCONV | 418.81 | 98.53 | 419.38 | 73.05 | 509.73 | 43.29 |
| + FINE-TUNE | 418.81 | 99.45 | 419.38 | 81.21 | 509.73 | 52.38 |

## A.5 VISUALIZING CLUSTERS

We verify the importance of MetaCluster qualitatively by plotting edge functions and the respective centroids associated with the first layer of FastKAN and MetaFastKAN. in Figures 4 and 5, respectively. From observing the functions associated with FastKAN and MetaFastKAN it is visually clear the functions associated with MetaFastKAN are much closer to the respective centroid than in FastKAN. Such a result demonstrates the importance using meta-learners to find a lower-dimensional functional space before clustering.

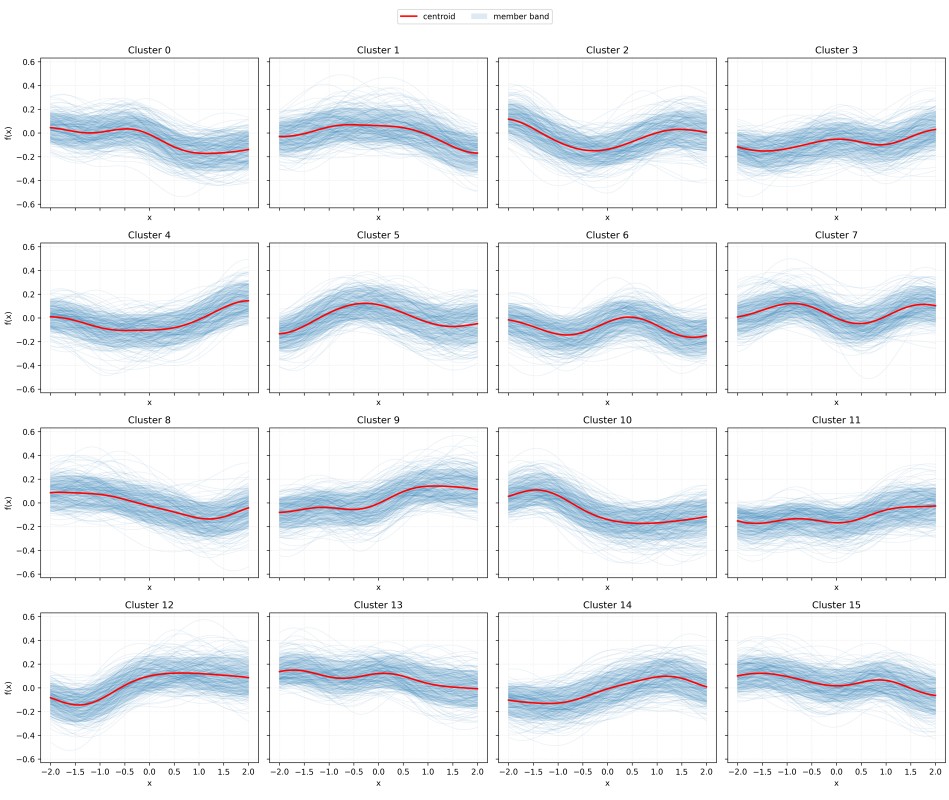

(a) The set of clusters showcasing the edge functions for each cluster on the first layer.

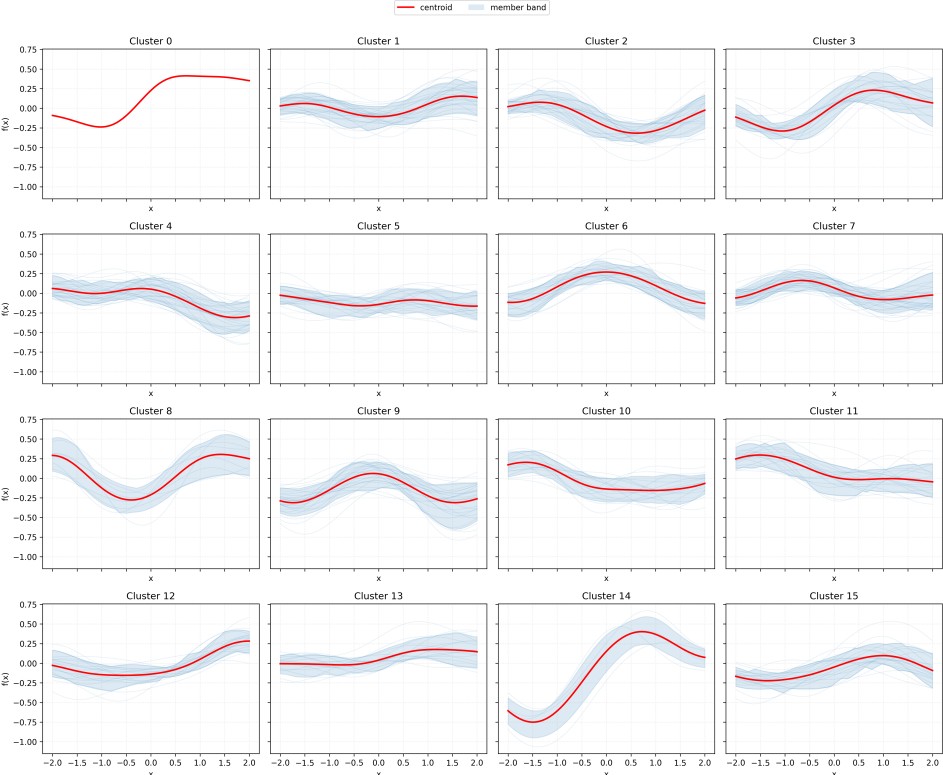

(b) The set of clusters showcasing the edge functions for each cluster on the second layer.

Figure 4: The edge functions for each cluster (in blue) and the centroid functions (in red) for the (a) first and (b) second layers of the fully-connected FastKAN models on Cifar-10.

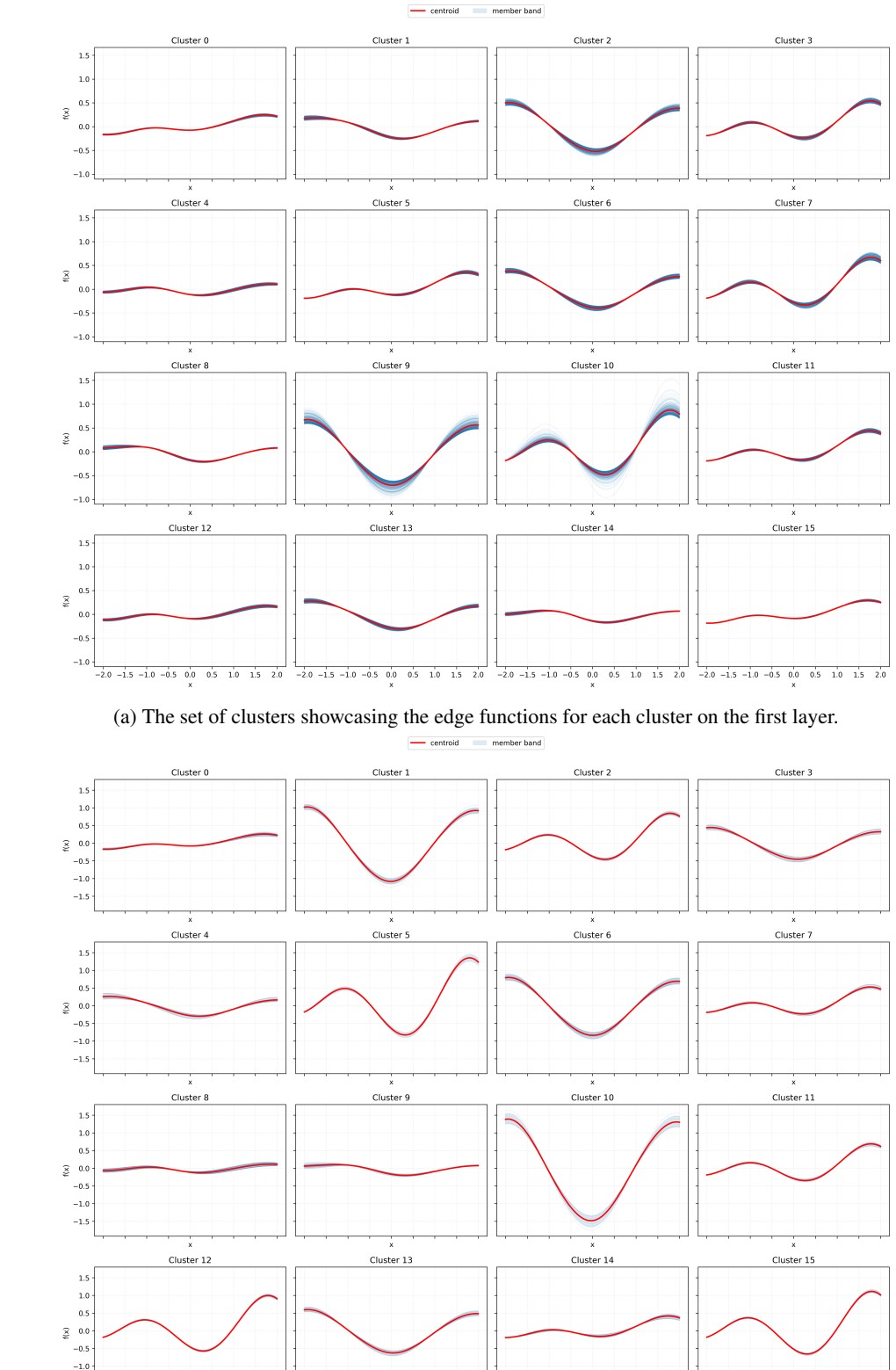

(a) The set of clusters showcasing the edge functions for each cluster on the first layer.

(b) The set of clusters showcasing the edge functions for each cluster on the second layer.

Figure 5: The edge functions for each cluster (blue) and the centroid functions (red) for the (a) first and (b) second layers of the fully-connected MetaFastKAN models on Cifar-10.

### A.6   LLM DISCLOSURE

We used large language models to assist in drafting and revising the manuscript.

