# OpenReview forum: "MetaCluster: Enabling Deep Compression of Kolmogorov-Arnold Network"
_ICLR.cc/2026/Conference — Submitted to ICLR 2026_

### Official Review · Reviewer_VcBJ · 2025-10-27

**Soundness:** 3
**Presentation:** 3
**Contribution:** 3
**Rating:** 6
**Confidence:** 4

**Summary:**

The paper addresses the memory overhead of Kolmogorov–Arnold Networks (KANs), where each edge carries a vector of basis coefficients rather than a scalar. It proposes MetaCluster, a three-stage pipeline: (1) train a lightweight meta-learner that maps low-dimensional embeddings to coefficient vectors so that coefficients lie on a clusterable low-dimensional manifold; (2) perform K-means in coefficient space and replace per-edge vectors with shared centroids plus indices; (3) discard the meta-learner and briefly fine-tune the centroid codebook. On MNIST, CIFAR-10/100, and both fully-connected KANs and ConvKANs with several basis families, MetaCluster reports up to 80× parameter-storage reduction without accuracy loss relative to the uncompressed KAN.

**Strengths:**

- Leveraging the vectorized per-edge structure of KANs makes codebook amortization particularly effective compared to scalar-weight MLPs. The method is simple to integrate: the meta-learner is used only during training and removed at inference.
- Clear articulation of why naïve weight sharing fails on KANs  and how manifold shaping mitigates this.
- Solid empirical coverage for small/medium scale: Results span FC-KAN and ConvKAN, with B-spline/RBF/Gram bases on MNIST/CIFAR-10/100.

**Weaknesses:**

- Experiments are limited to MNIST/CIFAR and relatively small KAN/ConvKANs; there is no large-scale vision or transformer-style model demonstration.
- The paper emphasizes zero inference overhead from removing the meta-learner, but provides no wall-clock or FLOPs comparison of training cost vs. baselines for the meta-learner + clustering + fine-tuning stages.
- While related work mentions Hessian-weighted K-means and differentiable K-means (DKM), the paper does not evaluate these variants or other clustering families (e.g., hierarchical or agglomerative).
- The paper uses a single global K per model family (e.g., FC-KAN K=16; ConvKAN K=256 per hyperparameter tables) and does not explore layer-wise varying K.

**Questions:**

1. Recent post-training, clustering-based methods (e.g., model folding[1], IFM[2] ) share the theme of parameter sharing/tying and low-dimensional structure. How does MetaCluster compare conceptually and empirically? Would it be possible to perform post-training clustering on a KAN model trained without metaclustering?

2. The authors state that quantization is complementary. Do you anticipate non-trivial accuracy loss when combining MetaCluster with 8-bit / 4-bit quantization of centroids and/or indices? Any preliminary data?

[1] Wang, Dong, et al. "Forget the data and fine-tuning! just fold the network to compress." arXiv preprint arXiv:2502.10216 (2025).
[2] Chen, Yiting, Zhanpeng Zhou, and Junchi Yan. "Going beyond neural network feature similarity: The network feature complexity and its interpretation using category theory." arXiv preprint arXiv:2310.06756 (2023).

---

> ### Author Response · Authors · 2025-12-03
>
> We appreciate the reviewer’s detailed feedback. We have addressed your main concerns through the following comments and revisions, and we hope these meet your expectations.
>
> > Experiments are limited to MNIST/CIFAR and relatively small KAN/ConvKANs; there is no large-scale vision or transformer-style model demonstration.
>
> Please see our explanation to eTyr.
>
> > The paper emphasizes zero inference overhead from removing the meta-learner, but provides no wall-clock or FLOPs comparison of training cost vs. baselines for the meta-learner + clustering + fine-tuning stages.
>
> Please see our explanation to Rvza.
>
> >While related work mentions Hessian-weighted K-means and differentiable K-means (DKM), the paper does not evaluate these variants or other clustering families (e.g., hierarchical or agglomerative).
>
> Please see our explanation to  Rvza for our reasoning for K-Means over alternative clustering algorithms.   We report Hessian-weighted K-means (GPTVQ) in Table 1.
>
> > The paper uses a single global K per model family (e.g., FC-KAN K=16; ConvKAN K=256 per hyperparameter tables) and does not explore layer-wise varying K.
>
> We appreciate this observation and agree that exploring layer-wise variation in K is an interesting direction for future work. In our current experiments, we adopted a single global K per model family (e.g., K=16 for fully connected KANs and K=256 for ConvKANs) primarily for simplicity and to ensure a fair comparison across architectures. These values were chosen based on the layer with the largest number of connections, which provides a conservative setting for compression.
> Our framework is inherently flexible and can accommodate layer-specific cluster counts without modification. Allowing K to vary by layer would likely amplify the benefits we already demonstrate: smaller layers could adopt fewer clusters without sacrificing accuracy, leading to even greater compression and reduced memory overhead beyond what we report. In other words, our results represent a strong baseline, and incorporating layer-wise K would only strengthen the case for MetaCluster’s effectiveness.
>
>
> > Recent post-training, clustering-based methods (e.g., model folding[1], IFM[2] ) share the theme of parameter sharing/tying and low-dimensional structure. How does MetaCluster compare conceptually and empirically?
>
> Conceptually, MetaCluster differs from post-training clustering approaches in that it actively shapes the weight space during training rather than clustering weights after they have been optimized. Post-training methods such as model folding and IFM assume that the trained weights already exhibit sufficient structure for clustering or folding, which might be true for scalar weights in MLPs or CNNs. However, KANs introduce a unique challenge: each edge carries a high-dimensional coefficient vector.  Furthermore, each edge is composed of nonlinear basis functions with strong within-edge dependencies.
>
> >Would it be possible to perform post-training clustering on a KAN model trained without metaclustering?
>
> We include results in the paper for post-training clustering on a KAN model trained without MetaCluster in Section 4.2. These results confirm the difficulty: clustering raw KAN weights without manifold shaping causes accuracy to collapse (e.g., CIFAR-10 accuracy drops from 47.52% to 27.92% for ClusterKAN), and even after fine-tuning, recovery remains incomplete (from 47.52% to 43.71% for ClusterKAN).
>
> > The authors state that quantization is complementary. Do you anticipate non-trivial accuracy loss when combining MetaCluster with 8-bit / 4-bit quantization of centroids and/or indices? Any preliminary data?
>
> We describe MetaCluster as complementary to traditional bit-width quantization because our framework operates at the level of weight sharing via clustering, which does not preclude applying quantization to the resulting centroids or indices. However, as with any compression technique, combining multiple methods introduces compounding sources of error. Therefore, we anticipate some additional accuracy degradation when applying 8-bit or 4-bit quantization on top of MetaCluster, particularly at very low bit-widths where quantization noise becomes significant.
>
> As the KAN community is relatively new, aside from QuantKAN [3], to our knowledge, no works have explored applying bit-width quantization to the KAN. As such, there is no universal bit-width quantization approach that can be coupled with MetaCluster.  Our focus in this work was to demonstrate the feasibility and effectiveness of clustering-based compression for KANs. Exploring hybrid approaches such as quantizing the centroid codebook after clustering is an important direction for future work, once quantization techniques for the KAN have matured.  We expect that careful calibration or fine-tuning could mitigate much of the accuracy loss.
>
> [3] Kazi Ahmed Asif Fuad and Lizhong Chen. Quantkan: A unified quantization framework for kolmogorov arnold networks

---

### Official Review · Reviewer_eTyr · 2025-10-31

**Soundness:** 3
**Presentation:** 3
**Contribution:** 3
**Rating:** 6
**Confidence:** 2

**Summary:**

The authors propose MetaCluster, a compression framework for Kolmogorov–Arnold Networks (KANs). Although KANs have demonstrated stronger performance than MLPs, they require significantly more parameters. This motivates a compression strategy based on weight sharing, where parameters are clustered into a small codebook, and only compact indices are stored. However, applying standard clustering directly to KANs is not straightforward. To address this, the authors train a small meta-learner that maps each edge’s coefficient vector onto a low-dimensional manifold, after which the vectors are clustered using K-means. The per-edge coefficients are then replaced by shared centroids (plus indices), followed by brief centroid fine-tuning. MetaCluster achieves up to an 80× reduction in parameter storage with no loss in accuracy.

**Strengths:**

The paper is well written, and its motivation is clear. The main strength lies in the impressive experimental results, as demonstrated in Tables 1 and 2. Additionally, the authors provide thorough ablation studies to validate their design choices, as shown in Section 4.3.

**Weaknesses:**

- The authors do not benchmark against non-KAN compression baselines. Given the extensive literature on model compression, it would be valuable to compare MetaCluster with common techniques (e.g., pruning, quantization, or weight sharing) applied to MLPs or CNNs. This would help clarify whether MetaCluster is state-of-the-art relative to general compression methods. If those methods are not easily extendable to KANs, a discussion explaining why would strengthen the paper.

- The evaluation is conducted only on relatively simple datasets (MNIST and CIFAR). It remains unclear how MetaCluster performs on more challenging or large-scale datasets.

- The meta-learner does induce a bit more of training complexity, since this adds engineering steps and hyperparameters (e.g., number of clusters, embedding dimnesions, etc) which can complicate the process. Can we also not jointly train but perhaps find a way to do post-training compression, i.e., learn a meta-learner afterwards? Perhaps decoupling these can ease the process a bit.

- The ablation results indicate that performance is sensitive to the meta-embedding dimension: clustering becomes more difficult as the embedding dimension increases. This suggests that finding a suitable dimension may require tuning and makes the method less plug-and-play.

**Questions:**

I have a few simple questions:

- Why choose K-means instead of other potential alternatives? Perhaps there are other choices that could boost the performance of MetaCluster.

- Does your compression ratio / memory account for codebook overhead + index storage + bit packing?

I apologize in advance if these were already answered in the manuscript and I missed them.

---

> ### Author Response · Authors · 2025-12-03
>
> We thank the reviewer for their thoughtful and comprehensive review. We appreciate the time and effort invested, and hope that our responses and revisions below address your main concerns.
>
> > The authors do not benchmark against non-KAN compression baselines. Given the extensive literature on model compression, it would be valuable to compare MetaCluster with common techniques (e.g., pruning, quantization, or weight sharing) applied to MLPs or CNNs. This would help clarify whether MetaCluster is state-of-the-art relative to general compression methods. If those methods are not easily extendable to KANs, a discussion explaining why would strengthen the paper.
>
> We appreciate this suggestion and agree that situating MetaCluster within the broader compression landscape is important. Our choice of weight sharing was deliberate: it directly addresses the dimensionality explosion introduced by KANs’ per-edge coefficient vectors, which is fundamentally different from the scalar weights in MLPs or CNNs. Techniques such as pruning or low-bit quantization, while highly effective for conventional architectures, do not fully exploit this structure and therefore offer limited benefit for KANs.
>
> To provide context from existing vector quantization literature, we include results for GPTVQ in Table 1. As shown, directly applying GPTVQ to KANs performs poorly and, in some cases, worse than standard vector quantization. This reinforces that methods designed for scalar-weight architectures do not translate effectively to KANs without significant adaptation.
>
> Regarding traditional bit-width quantization, we reference QuantKAN [1], which systematically benchmarks state-of-the-art quantization methods (LSQ, LSQ+, PACT, DoReFa, GPTQ, BRECQ, AWQ, HAWQ-V2) on KANs. Their results show that even aggressive low-bit quantization (e.g., 4-bit weights) achieves at most ~8x compression before severe accuracy degradation occurs. In contrast, MetaCluster achieves up to 80x compression with negligible accuracy loss.
>
> [1] Kazi Ahmed Asif Fuad and Lizhong Chen. Quantkan: A unified quantization framework for kolmogorov arnold networks.
>
> > The evaluation is conducted only on relatively simple datasets (MNIST and CIFAR). It remains unclear how MetaCluster performs on more challenging or large-scale datasets.
>
> We appreciate this observation. Our work is intended as a foundational step toward compressing KANs, which have only recently emerged and currently lack established large-scale benchmarks or compression baselines. We selected MNIST, CIFAR-10, and CIFAR-100 for three key reasons:
> 1. **Alignment with Prior KAN Research**:
> These datasets are standard in the KAN literature [1,2,3], enabling direct comparison and reproducibility. At present, large-scale benchmarks such as ImageNet are not widely adopted for KANs, making these datasets the most relevant for fair evaluation.
> 2. **Proof of Concept for High-Dimensional Compression**:
> Our primary goal was to demonstrate that MetaCluster can achieve up to 80x compression on image classification with negligible accuracy loss under controlled conditions. These datasets provide a well-understood baseline for validating the core idea before scaling to more complex domains.
> 3. **Scalability by Design**:
> MetaCluster introduces no architectural constraints that limit its applicability to larger datasets or models. Both the meta-learner and clustering steps scale linearly with the number of edges, and the codebook amortization becomes even more advantageous as model size grows. We anticipate that the benefits will be amplified for large-scale KANs, where memory overhead is a critical bottleneck.
>
> In summary, while our current evaluation focuses on widely used benchmarks for reproducibility, the proposed framework is inherently scalable and well-suited for future work on large-scale datasets such as ImageNet.
>
> [1] Zhangchi Zhao, Jun Shu, Deyu Meng, and Zongben Xu. Improving memory efficiency for training
> kans via meta learning
>
> [2] Kazi Ahmed Asif Fuad and Lizhong Chen. Quantkan: A unified quantization framework for kolmogorov arnold networks
>
> [3] Ivan Drokin. Kolmogorov-arnold convolutions: Design principles and empirical studies

---

> > ### Author Response · Authors · 2025-12-03
> >
> > > The meta-learner does induce a bit more of training complexity, since this adds engineering steps and hyperparameters (e.g., number of clusters, embedding dimensions, etc) which can complicate the process. Can we also not jointly train but perhaps find a way to do post-training compression, i.e., learn a meta-learner afterwards? Perhaps decoupling these can ease the process a bit.
> >
> > We acknowledge that introducing a meta-learner adds some training complexity due to additional engineering steps and hyperparameters (e.g., number of clusters, embedding dimensions). However, this design choice is intentional and beneficial. The meta-learner enables a substantial reduction in memory footprint during training and inference, which is critical for scaling Kolmogorov-Arnold Networks (KANs). Moreover, the extra hyperparameters provide flexibility rather than burden: by tuning cluster count and embedding dimension, users can trade off compression level against downstream performance to meet task-specific requirements.
> >
> > To address concerns about computational overhead, we provide quantitative results in Section 4.3. For a more thorough discussion, see our response to Rvza.
> >
> > Regarding the suggestion to decouple the meta-learner and apply it post-training, this is unfortunately not feasible for our approach. The meta-learner’s primary role is to shape KAN coefficient vectors onto a task-aligned low-dimensional manifold during optimization. This manifold structure is what makes clustering effective. If the meta-learner were trained after the fact, it would only observe fixed weights and could not influence their geometry during training, resulting in a poorly organized manifold and ineffective clustering. Joint optimization is therefore essential to achieve the reported compression and accuracy guarantees.
> >
> > >The ablation results indicate that performance is sensitive to the meta-embedding dimension: clustering becomes more difficult as the embedding dimension increases. This suggests that finding a suitable dimension may require tuning and makes the method less plug-and-play.
> >
> > We agree that the embedding dimension influences clustering quality, as shown in our ablation results. However, we view this not as a limitation but as a mechanism for controllable compression-accuracy trade-offs. The embedding dimension directly shapes the geometry of the coefficient manifold:
> > - Lower dimensions impose stronger regularization, making clustering easier and enabling higher compression at the cost of slightly reduced expressivity.
> > - Higher dimensions allow greater flexibility for accuracy but reduce clusterability.
> >
> > This tunable parameter gives practitioners explicit control over the balance between memory and performance, which is highly valuable in deployment scenarios with strict resource constraints. While extreme increases in embedding dimension degrade clustering quality, our experiments show that small dimensions (1–2) consistently deliver strong results across datasets and architectures. These settings are simple to adopt and require minimal tuning, making the method practical for most use cases.
> >
> > >Why choose K-means instead of other potential alternatives? Perhaps there are other choices that could boost the performance of MetaCluster.
> >
> > Please refer to our earlier explanation to Rvza.
> >
> > > Does your compression ratio / memory account for codebook overhead + index storage + bit packing?
> >
> > Yes, our compression ratio takes into account the codebook overhead+index storage + bit packing.

---

### Official Review · Reviewer_PLUa · 2025-11-01

**Soundness:** 3
**Presentation:** 4
**Contribution:** 3
**Rating:** 4
**Confidence:** 4

**Summary:**

This paper proposes MetaCluster, a three-stage compression framework for Kolmogorov-Arnold Networks (KANs) that combines meta-learning with weight sharing. In the first stage, a small meta-learner maps low-dimensional embeddings into coefficient vectors, constraining them to lie on a manifold that is highly clusterable. Then, K-means clustering is applied to replace per-edge coefficient vectors with shared codebook centroids. Finally, the centroids are fine-tuned to recover accuracy. The authors report up to 80× reduction in parameter storage with minimal accuracy loss across multiple architectures and datasets.

**Strengths:**

- The paper clearly identifies KAN’s memory inefficiency and proposes a targeted solution.
- This is the first successful application of weight sharing specifically designed for KANs.
- The paper is well organized and easy to follow.

**Weaknesses:**

- Complete absence of vector quantization literature. The proposed method is fundamentally vector quantization (VQ): mapping high-dimensional vectors to discrete codebook entries via clustering. However, the paper never mentions "vector quantization" and ignores relevant research.
- Lack of comparison with established vector quantization methods. The paper employs standard K-means with Euclidean distance but provides no comparison against advanced VQ techniques. For example, Product Quantization [1], which decomposes vectors into subvectors quantized independently, could achieve superior compression ratios. The choice of Euclidean distance over alternatives (cosine similarity, learned metrics) is also unjustified—for coefficient vectors representing basis functions, cosine similarity might better preserve functional shape. Without these comparisons, we cannot assess whether the meta-learner genuinely adds value over simpler VQ baselines.
- The paper motivates KAN compression with the references of KNN’s advantages in scientific tasks. However, all experiments are conducted on computer vision tasks (MNIST, CIFAR-10, CIFAR-100). This is largely different from the scientific tasks where KNN is explored. It would strengthen the paper to include evaluations on domains that better reflect the stated motivation, such as scientific or physical modeling tasks.

[1] Jégou et al. (2010) proposed product quantization for efficient nearest neighbor search.

**Questions:**

- Can you provide results on scientific computing tasks (equation modeling, PDE solving) where KANs have demonstrated their primary advantages, rather than only vision tasks?
- How does your method compare against Product Quantization, which decomposes vectors into independently quantized subvectors and could achieve smaller codebook sizes?

---

> ### Author Response · Authors · 2025-12-03
>
> We thank the reviewer for presenting their thoughtful and thorough review. We hope our comments and revisions below address your main concerns.
>
> > Complete absence of vector quantization literature.
>
> Thank you for pointing this out. Our method is conceptually related to vector quantization (VQ), as both approaches rely on codebook-based compression. In the revision in Section 5.3, we have added a dedicated discussion of this connection and cited key VQ works to provide proper context.
> While VQ offers the theoretical foundation, our contribution lies in adapting these principles to Kolmogorov-Arnold Networks and addressing challenges unique to this setting, such as meta-learned manifold shaping and clustering in high-dimensional basis function coefficient spaces, which are not covered in classical VQ literature.
>
> >Lack of comparison with established vector quantization methods. The paper employs standard K-means with Euclidean distance but provides no comparison against advanced VQ techniques. For example, Product Quantization [1], which decomposes vectors into subvectors quantized independently, could achieve superior compression ratios.
>
> Thank you for highlighting this point.  As mentioned in [2], which adapts [1], their VQ approach achieves compression by partitioning an mxn weight matrix into segments of size d, and clustering each segment independently. The resulting compression factor for a layer with d-dimensional subvectors is: $\frac{32mn}{32kn + \log_2(k)m\left(\frac{n}{d}\right)}$.
>
> This follows from:
>  - Original Size: $32mn$ bits
>  - Codebooks: For each of the n/d segments, store k cluster centers of size d: $kd\left(\frac{n}{d}\right)$
>  - Indices: For each row and segment, store an index requiring log(k) bits: $m\left(\frac{n}{d}\right)log(k)$
>
> When mapping this formulation to KANs, where each edge carries a vector of $|w|$ coefficients, the analogous expression becomes: $\frac{32mn|w|}{32k|w| + \log_2(k)m|w|\left(\frac{n}{d}\right)}$. Our approach achieves a higher compression factor because it shifts the bottleneck from the codebook size (which grows with k and $|w|$) to the index size, which scales more favorably.
>
> The only scenario where VQ decomposition like [2] might help is if we cluster centroids across multiple KAN edges, potentially reducing k and balancing the trade-off between the codebook and indices.  However, this would require discovering a lower-dimensional manifold for effective clustering, precisely what our meta-learning stage accomplishes. Without such manifold shaping, VQ suffers from the curse of dimensionality and poor cluster quality.
>
> For comparison with alternative VQ approaches, we include results for GPTVQ applied to the KAN in Table 1. Table 1 shows that traditional vector quantization methods based on Hessian-aware information are not effective with the KAN.
>
> [2] Yunchao Gong, Liu Liu, Ming Yang, and Lubomir Bourdev. Compressing deep convolutional networks using vector quantization.
>
> >The choice of Euclidean distance over alternatives (cosine similarity, learned metrics) is also unjustified—for coefficient vectors representing basis functions, cosine similarity might better preserve functional shape. Without these comparisons, we cannot assess whether the meta-learner genuinely adds value over simpler VQ baselines.
>
> Our use of Euclidean distance follows directly from the properties of the K-means algorithm. K-means minimizes the sum of squared Euclidean distances between points and their assigned centroids, and its update step relies on this metric to compute cluster means in closed form. Substituting a different metric (e.g., cosine similarity or a learned distance) would break this property and require a fundamentally different clustering procedure, such as K-medoids or metric-learning-based clustering, which introduces additional complexity and computational overhead.
>
> We qualitatively justify our use of Euclidean distance in Appendix A.5, where we can see with our current methodology that the functions of MetaFastKAN already lie near the centroids due to the reduced functional space, unlike FastKAN. These visualizations confirm that the manifold is sufficiently compact and well-aligned, making Euclidean distance not only effective but sufficient for capturing meaningful relationships.
>
> To address the concern quantitatively, we experimented with cosine similarity as an alternative metric. The results, included in Appendix A.3, show a slight decrease in downstream accuracy after compression compared to Euclidean distance. While cosine similarity emphasizes directional alignment, it ignores vector magnitudes, which appear to carry meaningful information in our setting. Euclidean distance, by contrast, leverages both direction and scale, making it better suited for our representation space. This advantage likely arises from the meta-learning stage, which organizes coefficient vectors into a structured manifold where absolute distances remain informative.

---

> > ### Author Response · Authors · 2025-12-03
> >
> > > Can you provide results on scientific computing tasks (equation modeling, PDE solving) where KANs have demonstrated their primary advantages, rather than only vision tasks?
> >
> > We agree that including results on equation modeling would strengthen the evaluation. Our initial focus was on vision tasks because these domains exhibit the largest memory footprint, making compression most impactful. In contrast, equation modeling tasks typically involve smaller models where memory is less of a bottleneck, so compression provides limited practical benefit.
> >
> > Nevertheless, to address your concern, we have added high-dimensional equation modeling results in Table 6, which achieve 124.1x compression with negligible changes in mean squared error. These experiments confirm that our approach maintains accuracy even in scientific computing settings, demonstrating that the proposed method generalizes beyond vision tasks.

---

### Official Review · Reviewer_Rvza · 2025-11-06

**Soundness:** 3
**Presentation:** 2
**Contribution:** 2
**Rating:** 4
**Confidence:** 3

**Summary:**

The authors propose a three stage weight sharing method for KAN. The first stage involves mapping low dimensional embeddings to per-edge coefficient vectors. The second stage involves k-means clustering to replace per-edge vectors with centroids. The final stage involves  finetuning centroid codebook to recover accuracy loss.

**Strengths:**

The proposed weight sharing method greatly reduces the amount of trainable parameters in KAN.

**Weaknesses:**

The proposed method crucially relies on k-means clustering to provide reasonable good centroids. However, k-means clustering assumes the data is spherically shaped, which may not be true in practice. Could the authors replace the K-means clustering by other clustering methods (e.g. gaussian mixture model) to illustrate the proposed method can be used together with different clustering algorithms?

**Questions:**

Could the authors theoretically quantify the accuracy loss from weight sharing, compared to vanilla KAN?
What is the complexity of the proposed 3-stage approach and how does it compare to vanilla KAN?
Could the authors report the standard error in experiments (table 1-3)? The standard error can potentially be obtained by using different train/validation splits.

---

> ### Author Response · Authors · 2025-12-03
>
> We would like to thank the reviewer for presenting their concerns.  We hope our comments and revisions below address your main concerns.
> > The proposed method crucially relies on k-means clustering to provide reasonable good centroids. However, k-means clustering assumes the data is spherically shaped, which may not be true in practice. Could the authors replace the K-means clustering by other clustering methods (e.g. gaussian mixture model) to illustrate the proposed method can be used together with different clustering algorithms?
>
> Our framework is compatible with alternative clustering algorithms. We chose K-means primarily because of its widespread use in weight-sharing literature for MLPs and its linear complexity with respect to the number of data points, which is critical for scalability.  While K-means assumes spherical clusters, its objective of minimizing within-cluster squared Euclidean distance aligns well with reducing quantization error in weight sharing. Other clustering methods often optimize different objectives, which may not directly serve this purpose.
>
> To address your suggestion, we have added experiments using Gaussian mixture models in Appendix A.4, demonstrating that MetaCluster remains effective when paired with alternative clustering strategies.
>
> > Could the authors theoretically quantify the accuracy loss from weight sharing, compared to vanilla KAN?
>
> We believe that a meaningful theoretical analysis is not possible because MetaCluster operates on a lower-dimensional manifold of KAN activations learned during training. This manifold shaping alters the weight distribution prior to clustering, making direct comparison with vanilla KAN weights infeasible.
>
> That said, clustering introduces quantization error, which we mitigate through fine-tuning. As shown in Tables 1, 2, and 6, fine-tuning effectively recovers accuracy, resulting in negligible performance degradation relative to the uncompressed baseline.
>
> >What is the complexity of the proposed 3-stage approach, and how does it compare to vanilla KAN?
>
> Our three-stage framework, consisting of training, clustering, and fine-tuning, introduces only minor computational overhead compared to vanilla KAN. The clustering stage is negligible (e.g., 0.029 s vs. 2.155 s per epoch), but the fine-tuning stage accounts for most of the additional cost, as it involves extra epochs after initial training. However, fine-tuning runs for far fewer epochs than the main training stage, so the overall increase remains minor. Importantly, inference performance is unchanged. MetaClusterKAN and MetaClusterKANConv achieve inference times that are on par with vanilla KAN counterparts, despite leveraging a codebook-based representation. For example, on CIFAR-10, inference for MetaClusterKAN is 0.521 s vs. 0.510 s for KAN, and similar trends hold for convolutional variants. In summary, our method achieves compression with minimal training overhead and no inference penalty, making it practical for real-world deployment. Full results are reported in Section 4.3
>
>
> Moreover, the slight increase in training cost is offset by a substantial reduction in memory footprint during and after training. As shown in Tables 1 and 2, our method achieves compression factors of up to 7.4x during training and 80x post-training, all without sacrificing accuracy.
>
> > Could the authors report the standard error in experiments (table 1-3)?
>
> Yes, we now include standard error bars in Tables 1-3.  These are computed from the results of 5 independent runs.

---

### Author Response · Authors · 2025-12-03

We appreciate the reviewers' time and effort in reviewing our paper. In response to their feedback and suggestions, we have submitted a revised version of the manuscript. Please refer to our individual responses to the reviewers for details on how we addressed their comments.

---

### Meta-Review · Area_Chair_KFtN · 2025-12-04

**Summary:**

Reviewer Rvza and PLUa each gave a score of 4, while reviewers VcBJ and eTyr gave a score of 6. The primary concerns raised by the reviewers focus on the experimental section—including whether the empirical evaluation sufficiently covers scientific experiments, whether ImageNet-level validation is included, and whether comparisons against more advanced or recent baselines are provided. With the exception of the ImageNet experiment, the authors have attempted to address these concerns in the rebuttal and revised draft, and their clarifications have resolved several of the reviewers’ questions. Given that the KAN lies in an emerging area, but the experimental scope should still be largely strengthened in their future work.

**Reviewer Concerns:**

The authors are able to respond to the concerns raised by reviewers Rvza and PLUa in a convincing and substantive manner with mathematical analysis and experiments.

**Reviewer Scores:**

From a problem-solving perspective, it is unlikely that Reviewer Rvza would significantly raise their score, as their core concern—whether the authors can theoretically quantify the accuracy loss introduced by weight sharing compared with the original KAN—remains only partially addressed. For Reviewer PLUa, the situation is similar: the request for an ImageNet-scale evaluation was not fully satisfied, and therefore a substantial score increase may not occur. However, I believe the overall impact of these concerns should be viewed in context. First, the authors provide a thorough theoretical analysis supporting the proposed KAN compression method, which directly addresses the motivation and design principles behind the approach. Second, ImageNet evaluation has not been a consistent requirement in prior KAN-related work, and its absence here should not be over-penalized.

---

### Decision · Program_Chairs · 2026-01-26

Reject